# Most Attractive Scenic Sites of the Bulgarian Black Sea Coast: Characterization and Sensitivity to Natural and Human Factors

**Alexis Mooser** [1,2], **Giorgio Anfuso** [2,*], **Hristo Stanchev** [3], **Margarita Stancheva** [3], **Allan T. Williams** [4] **and Pietro P. C. Aucelli** [1]

1   Department of Science and Technology (DiST), Parthenope University, 80143 Naples, Italy; alex.moosr@gmail.com (A.M.); pietro.aucelli@uniparthenope.it (P.P.C.A.)
2   Faculty of Marine and Environmental Sciences, University of Cádiz, Polígono Río San Pedro s/n, 11510 Puerto Real, Spain
3   Center for Coastal and Marine Studies (CCMS), 9000 Varna, Bulgaria; stanchev@ccms.bg (H.S.); stancheva@ccms.bg (M.S.)
4   Department of Architecture, Computing and Engineering, University of Wales: Trinity Saint David (Swansea), Mount Pleasant, Swansea SA1 6ED, UK; allanwilliams512@outlook.com
*   Correspondence: giorgio.anfuso@uca.es

**Abstract:** Beach management is a complex process that demands a multidisciplinary approach, as beaches display a large variety of functions, e.g., protection, recreation and associated biodiversity conservation. Frequently, conflicts of interest arise, since management approaches are usually focused on recreation, preferring short-term benefits over sustainable development strategies; meanwhile, coastal areas have to adapt and face a changing environment under the effects of long-term climate change. Based on a "Sea, Sun and Sand (3S)" market, coastal tourism has become a major economic sector that depends completely on the coastal ecosystem quality, whilst strongly contributing to its deterioration by putting at risk its sustainability. Among beach users' preferences, five parameters stand out: safety, facilities, water quality, litter and scenery (the "Big Five"), and the latter is the focus of this paper. Bulgaria has impressive scenic diversity and uniqueness, presenting real challenges and opportunities as an emerging tourist destination in terms of sustainable development. However, most developing countries tend to ignore mistakes made previously by developed ones. In this paper, scenic beauty at 16 coastal sites was field-tested by using a well-known methodology, i.e., the Coastal Scenic Evaluation System (CSES), which enables the calculation of an Evaluation Index "D" based on 26 physical and human parameters, utilizing fuzzy logic matrices. An assessment was made of these high-quality sites located in Burgas (8), Varna (3) and Dobrich (4) provinces. Their sensitivity to natural processes (in a climate change context) and human pressure (considering tourist trends and population increases at the municipality scale) were quantified via the Coastal Scenic Sensitivity Indexes (CSSIs) method. The CSES and CSSI methods allowed us to conduct site classification within different scenic categories, reflecting their attractiveness (Classes I–V; CSES) and level of sensitivity (Groups I–III; CSSI). Their relationship made it possible to identify management priorities: the main scenic impacts and sensitivity issues were analyzed in detail and characterized, and judicious measures were proposed for the scenic preservation and enhancement of the investigated sites. Seven sites were classified as extremely attractive (Class I; CSES), but with slight management efforts; several Class II sites could be upgraded as top scenic sites, e.g., by cleaning and monitoring beach litter. This paper also reveals that investigated sectors were more sensitive to environmental impacts than human pressure; for example, eight were categorized as being very sensitive to natural processes (Group III; CSSI).

**Keywords:** landscape; beach; management; climate change; erosion; tourism pressure; sustainability; developing country

## 1. Introduction

Coastal areas host relevant aquatic and terrestrial ecosystems located at the interface between water and land [1] and have an intrinsic environmental value due to their great biodiversity that supports the provision of several ecosystem services and related functions essential for human subsistence [2,3]. Recreational and cultural activities are also relevant in coastal areas from several decades [4]. Such sensible and valuable environments are often threatened by natural processes and, in past decades, by an increasing level of population and human pressure [5–9].

Natural processes, such as coastal erosion and flooding, are often exacerbated by human-related activities [10] and linked to chronic erosion processes [11] and/or the impact of very energetic events, such as storms and hurricanes [12,13]. They constitute a rising issue enhanced by sea-level rise and other climatic-change-related processes, such as the increasing height of extreme waves, or a change in wave tracks, the intensity/frequency of storms and hurricanes [13–16]. Coastal sensitivity is the susceptibility of coastal environments to be affected by either inundation or erosion processes, and many studies have shown that over 70% of global shorelines are currently retreating because of climate change's processes [17]. This is a relevant problem that affects the majority of global coastal areas and is reflected in the reduction or complete loss of beach and dune systems and other relevant coastal environments, such as salt marshes and mangrove swamps [18,19]. Lincke and Hinkel [20] projected that, by 2100, there will be a global coastal land loss of around 60,000–415,000 km$^2$ and associated migration of 17–72 million people. The total or partial degradation of such environments would involve a loss of associated tourist, aesthetic and natural values [21–24]. Such a trend is emphasized when landward migration of coastal ecosystems is impeded because of the presence of seawalls or human settlements [25], a process known as "coastal squeeze" [26,27].

The population is expected to rise, and projections postulate an increase from 625 million (in 2000) up to 949 and 1388 million people in 2030 and 2060, respectively [28]. Infrastructure and activities related to human developments (tourist, fishing, industrial, etc.) therefore represent a worrying and increasing threat to coastal environments [28,29]. Land alteration is one of the most critical issues to manage, contributing to coastal erosion; land fragmentation; loss of habitats, biodiversity and ecosystem services; and landscape degradation. As an example, benefits derived from ecosystem services for the Mediterranean Sea, which represents 0.82% of the global ocean surface, are estimated to be over €26 billion a year [30]. Despite the coastal population carrying out a broad range of economic and environmental activities, in past decades, the increase of population and development has been greatly related to tourism, which is one of the fastest-growing industries worldwide: in 2019, it generated 10.3% of global Gross Domestic Product (GDP) and supported 330 million jobs [31], with coastal and marine tourism being the largest segment of this industry [32]. This is mainly due to the attraction of the "Sea, Sun and Sand" ("3S") tourism [33,34], which mostly characterized by peaks of visits limited to the summer season, due to a strong dependence on local weather conditions and partially to the coincidence of long breaks in schools, firms, etc. [8]. This can lead to overcrowded scenarios, a situation particularly affecting a site's scenic attractiveness and its associated prestige and positive image. Commonly, conflicts and adverse effects on the environment are caused by a lack of knowledge and weak management strategies, which frequently are non-existent. Coastal visitors are especially interested in beaches, a market that annually involves billions of US dollars [35]. This raised the following question: what are the visitors' preferences on beach choice? Research indicated five main parameters, the "Big Five", i.e., safety, no litter, water quality, facilities and scenery [33], and the latter, which is the main concern of this paper, is described as "an area, as perceived by people, whose character is the result of the action and interaction of natural and/or human factors" [36].

In this paper, the most attractive scenic sites of the Black Sea Bulgarian coast have been characterized and their sensitivity to climate-change-related processes and human pressure analyzed according to the methodology proposed by Mooser et al. [37,38]. The Coastal

Scenery Evaluation System (CSES) [39–41] was used to characterize the most attractive coastal scenic sites along the investigated area, together with the Coastal Scenic Sensitivity Index (CSSI) [37] to determine present and future coastal scenic sites' sensitivity to both natural processes and human interventions. Both are linked to the increasing coastal development associated with tourist demand in Bulgaria, where tourist arrivals grew in 2018 at a rate of 8% [42]. The results obtained in this paper constitute a useful tool for the preservation and enhancement of coastal beauty of the investigated areas and provide a basis for any sound foreseeable development plan devoted to landscape conservation.

## 2. Study Area

Bulgaria, which covers an area of 110,842 km², is situated on the Balkan Peninsula in Southeast Europe and has a 432 km in length microtidal coastline, which faces the western part of the Black Sea between Cape Rezovo to the south (at the border with Turkey) and Cape Sivriburun to the north (at the border with Romania) [43]. Most rivers flowing into the Black Sea along the Bulgarian coast are small, except for the Kamchia River. The coast comprises a variety of geomorphological features, showing a wide variety of coastal scenery: rock cliffs, sandy beaches, low-lying areas with estuaries and lagoons. The location of the investigated sites is presented in Figure 1.

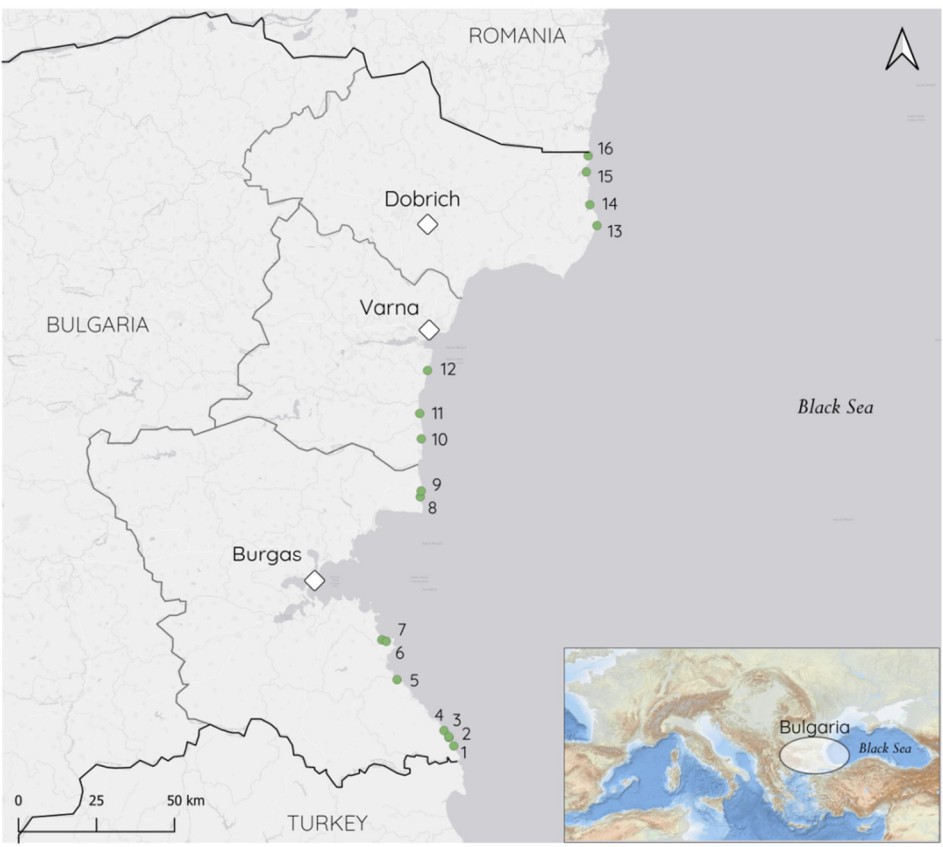

**Figure 1.** Location map of the 16 investigated sites: (1) Silistar, (2) Lipite, (3) Lipite Pocket Beach (PB), (4) Veleka, (5) Koral, (6) Ropotamo, (7) Arkutino, (8) Irakli, (9) Vaya PB, (10) Karadere, (11) Kamchia, (12) Rakitnika, (13) Cape Shabla Shore Platform (SP), (14) Shabla–Ezerets Lakes, (15) Durankulak Lake and (16) Durankulak North.

Cliffs are the most common feature, covering 49.3%, or 213 km, of the whole shoreline. Sand beaches constitute 34.5% (149 km) of the coast and the armored/engineered coast occupies 16.2% (70 km) [43]. Beach erosion and cliff retreat, both natural and human-induced, are the main hazards affecting the coastline [44]. Such retreat is partially linked to sea-level rise that, along the Bulgarian Black Sea coast, varies from 1.5 to 3 mm/y [45].

Around 20% of the Bulgarian coast has been identified as vulnerable to inundation at given scenarios of sea-level rise (0–5 m) [46]. Coastal storms, which acquire great relevance in coastal erosion, are extreme meteorological events that mainly occur in winter and are associated with severe N and NE winds.

In the recent past, the Bulgarian coast was covered by large dune systems that, despite being protected environmental areas, have experienced relevant reductions because of human activities and development; today, they occupy only 10% of the entire country's coastline [47]. The existing diversity of coastline features/landscapes makes the Bulgarian coast a popular destination for both homes of local people and accommodation for domestic and foreign tourists. Further, the coast shows favorable natural conditions for the seaside tourism development, as a result of a temperate climate and wide beaches with fine-grained sand. In the southern part of the country, pocket beaches are quite common, while in the central and northern parts, large sand beaches are common.

There are 14 (out of 264) coastal municipalities occupying an area of 5770 km$^2$, corresponding to 5.2% of the entire country's territory, that accommodates 726,923 residents, i.e., 9.8% of the national population according to the 2011 Census data [48].

Heavily concentrated in the Black Sea coast, tourism is a central pillar of the Bulgarian economy: in 2018, it formed 10.4% of the GDP (€6.46 billion) and provided a total of 346,800 jobs. Coastal tourism is today the most significant subsector and the fastest-growing part of the local economy involving almost 2/3 of the tourist infrastructure and tourists who mostly relate to beach-based activities, as in the "3S" tourism [49]. It contributed 66% to the Blue economy jobs (48,300 persons employed) and 55% to the Gross Value Added (or €399 million) in 2017 [50]. The first large sea resorts were established during the 1950s and 1960s. The most significant influence from coastal tourism development began at the end of the 1990s and has been expanding steadily since 2005 [47]. In 2018, Bulgaria accommodated more than 12 million international visitors, which increased by over one million during the 2012–2018 period, with half originating from the EU. Tourism is essential to many local economies, but to preserve such economic benefits, it is mandatory to soundly manage destinations and to conserve the natural aspects in which tourists are interested: this is the challenge for coastal managers in the 21st century [33,49].

Finally, cultural heritage is intrinsically connected to Bulgaria as a Black Sea country. It is part of its history, daily life, culture and tourism. Eleven Bulgarian sites are included in the UNESCO list of tangible and intangible world cultural and natural heritage. The coast is also an archaeology important area, where numerous underwater and coastal archaeological sites from different periods have been discovered—Prehistory, Antiquity (ancient Greek, Hellenistic and Roman), Mediaeval (Early Byzantium and Bulgarian). This rich concentration of submerged sites provides a worldwide unique archive of data to investigate social and economic aspects of ancient civilizations and cultures, but today such remains are at risk because of intensive human activities, e.g., fishing, hydrocarbon exploration, dredging of ports, etc. [51,52].

## 3. Methods

Along the Black Sea Bulgarian coast, 16 of the most attractive coastal scenic sites were evaluated by using the Coastal Scenic Evaluation System (CSES) [39–41] and the Coastal Scenic Sensitivity Index (CSSI) [37,38], as sites can potentially be affected in future decades by climate-change-related processes and increased tourist pressure. The key aspects of both methods, as stated in Mooser et al. [37,38], are presented in Figure 2.

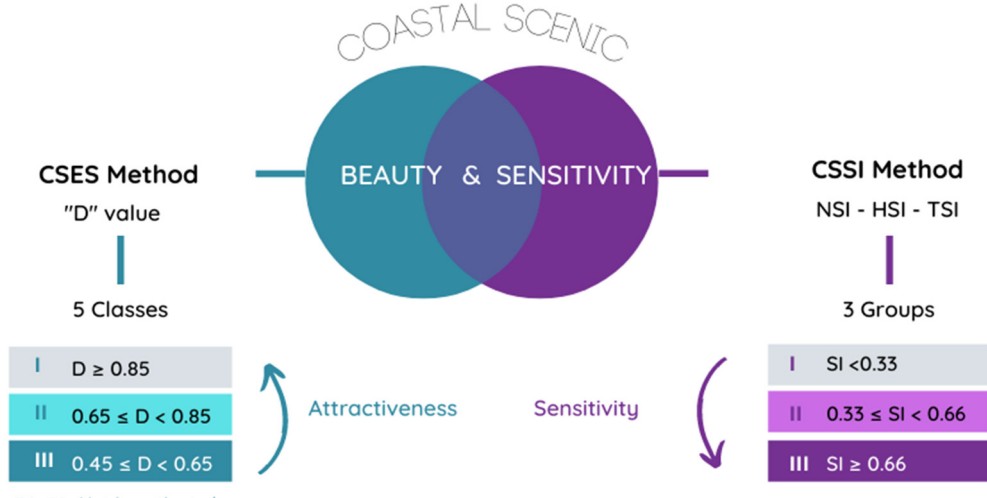

**Figure 2.** Summary of methods: indexes, scenic classes (Coastal Scenic Evaluation System; CSES) and sensitive groups (Coastal Scenic Sensitivity Indexes, CSSI).

**Scenic beauty assessment**. The focus of this paper is the research of Class I and II, sites and, secondary, Class III sites (Figure 2), that are accessible for beach users by a walk <1.5 h from the nearest car parking. The selection process was realized according to the same standard applied in the Balearic Islands and Andalusia, Spain [38,53], as detailed in the following lines:

(i) A first approximation on the location of the most attractive coastal areas was obtained via land-cover viewers and satellite images, e.g., Google Earth, Copernicus land viewer, etc. The images were used to eliminate urban and village areas, and preselect areas that appeared of great scenic values conforming to the 26 physical and human parameters (Table 1; CSES), e.g., a site which shoreline consists of cliff formations, extensive dunes system and/or shows high vegetation cover and, at the same time, records a low visual impact of human activities. If doubts arose relating the access difficulty or attractiveness, locations were automatically preselected.

(ii) Consequently, 26 sites respectively located in remote areas (17) and rural areas (9), were initially chosen after image viewing, irrespective as to whether they were located or not in protected areas.

(iii) After discussion with local coastal experts and detailed investigation of preselected areas, e.g., by consulting official webs of tourism and location of protected areas, review of published papers and grey literature, etc., a total of 21 locations were selected for field surveying. It should be noted that the criteria used to determine the distribution of preselected sites ranged according to the scenic variety of the shoreline investigated. Selected sites presented the greatest spatial density along heterogeneous scenic shorelines, such as Burgas province (points 1–4, Figure 1), whilst the opposite was true for homogenous scenic shorelines, e.g., Kamchia or Durankulak (points 11 and 15, Figure 1).

(iv) Field surveys were carried out in June 2021 between 10 a.m. and 6 p.m., during normal weather conditions, when stable conditions ruled (e.g., a storm affects water color, point 16 in Table 1) and over beach sectors 400–500 m in length; that is, when a long shoreline is assessed, it can be divided into different 400–500 m sectors. A few preselected sites such as Butamyata (Sinemorets) or Blatoto Alepu (also known as drivers' beach; Primorsko) were visited but, finally, not chosen, because of their low scenic quality. Cape Emine was finally not assessed because of the very regrettable condition of the access pathway, which required a walk estimated > 1.5 h. When constant alongshore scenic conditions were observed, adjacent sectors were joined together [54,55], giving finally a total of 16 coastal sites with different coastal lengths,

from a 62 m in length pocket beach, i.e., Lipite PB (point 2, Figure 1), to a 6770 m long beach, i.e., Durankulak (point 15, Figure 1), and covering a total of c. 32 km, i.e., 8% of the total coastal length of Bulgaria.

**Table 1.** Coastal Scenic Evaluation System (CSES) parameters with their corresponding weight and attribute scale. * Cliff special features: indentation, banding, folding, screes and irregular profile. ** Coastal landscape features: Peninsulas, rock ridges, irregular headlands, arches, windows, caves, waterfalls, deltas, lagoons, islands, stacks, estuaries, reefs, fauna, embayment, tombola, etc. *** Utilities: power lines, pipelines, street lamps, groins, seawalls, revetments, restaurants, etc.

| No. | Physical Parameters | | Weight | Rating | | | | |
|---|---|---|---|---|---|---|---|---|
| | | | | 1 | 2 | 3 | 4 | 5 |
| 1 | | Height (m) | 0.02 | Absent | $5 \leq H < 30$ | $30 \leq H < 60$ | $60 \leq H < 90$ | $H \geq 90$ |
| 2 | CLIFF | Slope | 0.02 | <45° | 45–60° | 60–75° | 75–85° | circa vertical |
| 3 | | Features * | 0.03 | Absent | 1 | 2 | 3 | Many (>3) |
| 4 | BEACH FACE | Type | 0.03 | Absent | Mud | Cobble/Boulder | Pebble/Gravel | Sand |
| 5 | | Width (m) | 0.03 | Absent | $W < 5$ or $W > 100$ | $5 \leq W < 25$ | $25 \leq W < 50$ | $50 \leq W \leq 100$ |
| 6 | | Color | 0.02 | Absent | Dark | Dark tan | Light tan/bleached | White/gold |
| 7 | ROCKY SHORE | Slope | 0.01 | Absent | <5° | 5–10° | 10–20° | 20–45° |
| 8 | | Extent | 0.01 | Absent | <5 m | 5–10 m | 10–20 m | >20 m |
| 9 | | Roughness | 0.02 | Absent | Distinctly jagged | Deeply pitted and/or irregular | Shallow pitted | Smooth |
| 10 | DUNES | | 0.04 | Absent | Remnants | Fore-dune | Secondary ridge | Several |
| 11 | VALLEY | | 0.08 | Absent | Dry valley | (<1 m) Stream | (1–4 m) Stream | River/limestone gorge |
| 12 | SKYLINE LANDFORM | | 0.08 | Not visible | Flat | Undulating | Highly undulating | Mountainous |
| 13 | TIDES | | 0.04 | Macro (>4 m) | | Meso (2–4 m) | | Micro (<2 m) |
| 14 | COASTAL LANDSCAPE FEATURES ** | | 0.12 | None | 1 | 2 | 3 | >3 |
| 15 | VISTAS | | 0.09 | Open on one side | Open on two sides | | Open on three sides | Open on four sides |
| 16 | WATER COLOR and CLARITY | | 0.14 | Muddy brown/grey | Milky blue/green | Green/grey/blue | Clear/dark blue | Very clear turquoise |
| 17 | NATURAL VEGETATION COVER | | 0.12 | Bare (<10% vegetation) | Scrub/garigue (marram, gorse) | Wetlands/meadow | Coppices, maquis (±mature trees) | Variety of mature trees |
| 18 | VEGETATION DEBRIS | | 0.09 | Continuous (>50 cm high) | Full strand line | Single accumulation | Few scattered items | None |
| | Human Parameters | | | | | | | |
| 19 | NOISE DISTURBANCE | | 0.14 | Intolerable | Tolerable | | Little | None |
| 20 | LITTER | | 0.15 | Continuous accumulations | Full strand line | Single accumulation | Few scattered items | Virtually absent |
| 21 | SEWAGE DISCHARGE EVIDENCE | | 0.15 | Sewage evidence | | Same evidence (1–3 items) | | No evidence of sewage |
| 22 | NON-BUILT ENVIRONMENT | | 0.06 | None | | Hedgerow/terracing/monoculture | | mixed cultivation ± trees/natural |
| 23 | BUILT ENVIRONMENT | | 0.14 | Heavy Industry | Heavy tourism and/or urban | Light tourism and/or urban | Sensitive tourism and/or urban | Historic and/or none |
| 24 | ACCESS TYPE | | 0.09 | No buffer zone/heavy traffic | No buffer zone/light traffic | | Parking lot visible from coastal area | Parking lot not visible from coastal area |
| 25 | SKYLINE | | 0.14 | Very unattractive | | Sensitively designed high/low | Very sensitively designed | Natural/historic features |
| 26 | UTILITIES *** | | 0.14 | >3 | 3 | 2 | 1 | None |

As previously stated, the Coastal Scenic Evaluation System (CSES) [39–41] is a method based on the evaluation of 26 parameters, 18 of which are physical and eight are human (Table 1). Such parameters were selected according to the results of numerous interviews of beach visitors in Turkey, Malta, Croatia, Portugal and the UK [39–41], and after discussion

between coastal experts, rated on a five-point attribute scale, with 1 indicating "absence" or "poor" quality and 5 "excellent/outstanding" quality (Table 1). Each parameter had a different weight (Table 1); that is, not all parameters are the same, but the weighting of all physical components is equal to that of the human parameters. The method is based on fuzzy logic mathematics and parameter weighting matrices, which allow one "to overcome subjectivity and quantify uncertainties" [39]. Fuzzy Logic Assessment (FLA) [56] is a scientific approach used to limit any mistake that the scenic value assessor makes. The assessor must tick one box for each parameter on the checklist (Table 1) and could tick the wrong attribute box (see corrections coefficients in Table 1). After fieldwork, results were presented as follows:

(i) Histograms, which provided a visual summary of both physical and human parameters obtained from Table 1 scores;
(ii) A weighted average of attributes, which delineated relative comparison of physical and human parameters;
(iii) Membership degree of attributes, which represented overall scenic assessment over the attributes.

All the above allow for the calculation of a scenic evaluation value "D" for each site that, according to the "D" value obtained, is categorized into five distinct classes (Figure 2), from Class I, i.e., extremely attractive natural sites with very high landscape values (D ≥ 0.85), to Class V, i.e., very unattractive urban sites with intensive development (D ≤ 0.0; see Anfuso et al. [57] for a detailed description and > 1000 study cases around the world).

**Scenic sensitivity assessment**. Present and future coastal scenic sites' sensitivity to natural processes and human interventions were obtained by using the Coastal Scenic Sensitivity Index (CSSI) [37,38] (Figures 2 and 3). The method allows for the determination of the intrinsic sensitivity of coastal scenic parameters as follows:

(i) Erosion/flooding processes in a climate change context,
(ii) Unsustainable coastal population and level of development—very often linked to the tourism industry and the lack of management.

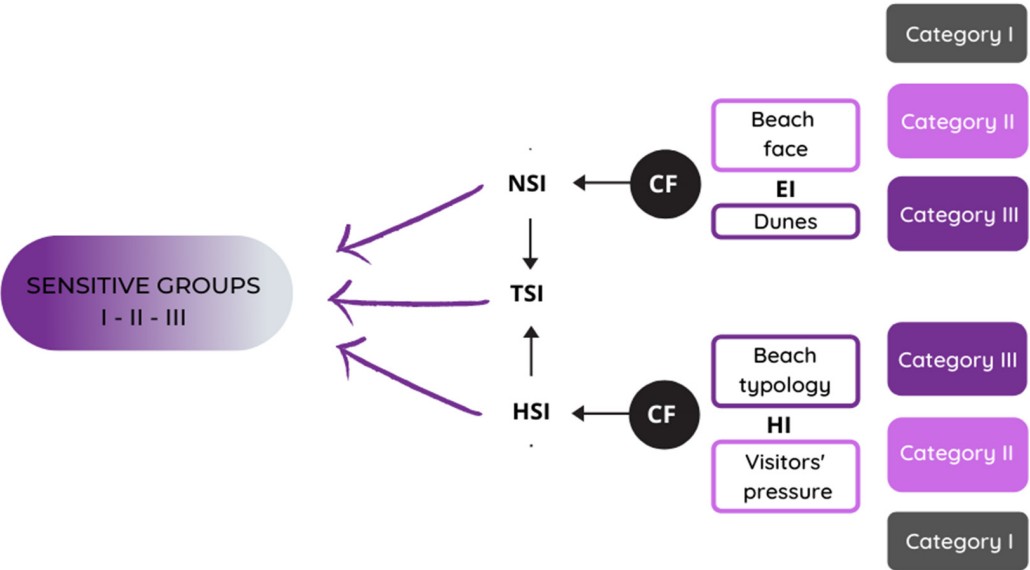

**Figure 3.** CSSI method: an overview of steps and parameters used for Natural Sensitivity Index (NSI), Human Sensitivity Index (HSI) and Total Sensitivity Index (TSI) assessment.

Concerning physical parameter sensitivity, Mooser et al. [37] considered that "Beach face" and "Dune" parameters (Table 2) were the most sensitive natural features to erosion/flooding processes that favor beach width reduction, i.e., low score at point 5 in

Table 1, sediment coarsening and darkening (points 4 and 6, Table 1) and dune erosion/disappearance (point 10, Table 1). Thereafter, the investigated areas were assessed according to the following standard (Figure 3):

(i)   During the first phase, sites were classified within three categories according to the presence/absence of the abovementioned parameters (Figure 3);

(ii)  In a second phase, the level of sensitivity of each site was determined by utilizing the Erodibility Index (EI) [37], which considers beach face and dune characteristics on a 1–5 scale (Table 2);

(iii) In a third phase, forcing variables and predicted changes of sea-level rise and storm surge were used to calculate a Correction Factor (CF, Table 2), since the effects of forcing factors on coastal environments are affected by future variations of those two variables;

(iv)  In a fourth phase, the final sensitivity of natural parameters, i.e., the Natural Sensitivity Index (NSI), was obtained by considering all the above in a 0–1 range of values (Table 2 and Figure 3), allowing for the categorization of the sites into three sensitive groups (Table 2).

**Table 2.** Erodibility Index (EI) parameters and Correction Factors (CFs) used for NSI assessment. * Only for Category III sites. ** Imminent Collapse Zone. *** Estimation expected by the end of the 21st century.

| Indexes and CF | | | Parameter | Null/Very Low (1) | Low (2) | Medium (3) | High (4) | Very High (5) |
|---|---|---|---|---|---|---|---|---|
| Natural Sensitivity Index | Erodibility Index | Beach face | Dry beach as a multiple of the ICZ ** | Accretion/ >5 times ICZ | 4 times ICZ | 3 times ICZ | 2 times ICZ | ≤ ICZ |
| | | | Sediment grain size | Gravel/pebbles | | Medium/ coarse sand or mixed | | Fine sand |
| | | | Rocky shore — Width | >80 | 80–60 | 60–40 | 40–20 | <20 |
| | | | Rocky shore — Location | Nearshore | | Foreshore | | Absent |
| | | Dunes * | Dune height (m) | ≥6 | ≥3 | ≥2 | ≥1 | <1 or absent |
| | | | Dune width (m) | >100 | >75 | >50 | >25 | <25 |
| | | | Vegetation cover | Complete with fixed dune (forest) | Complete with fixed dune (shrub) | Semi-complete (without fixed dune) | Semi-completed (without embryo dune) | Incomplete or absent |
| | | | Washovers (%) | 0 | ≤5 | ≤25 | ≤50 | ≥50 |
| | Correction Factor | Forcing | Significant wave height (m) | <0.75 | | 0.75–1.5 | | >1.5 |
| | | | Angle of approach | 10°–45° (Oblique) | | 0°–10° (Sub-parallel) | | 0° (Parallel) |
| | | | Tidal range | Macro tidal | | Meso tidal | | Micro tidal |
| | | Trends | Sea level rise (cm) *** | <0 | | 0–40 | | >40 |
| | | | Storm surge (m) *** | <1.5 | | 1.5–3 | | >3 |

Concerning the sensitivity of human parameters, Mooser et al. [37] suggested that "Noise disturbance", "Litter" and "Sewage discharge evidence" (points 19–21, Table 1) were essentially linked to the influence of beach visitors; meanwhile, parameters such as "Non-built environment", "Built environment", "Access type" and "Utilities" (points 22–24 and 26, Table 1) were principally related to the site protection feature (if any); and "Skyline" (point 25, Table 1) was used to the urbanization level of the surrounding areas. All the above essentially depend on land use and beach typology [37]. Therefore, we employed the following:

(i)   In a first phase, according to the level and typology of human pressure, each site was classified within one of the three pre-established categories (Figure 3);

(ii)  In a second phase, "Visitors pressure" and "Beach typology" (Table 3) were determined on a 1–5 scale, and the Human Impact Index (HI, Table 3 and Figure 3) was calculated;

(iii) In a third phase, a Correction Factor (CF, Table 3) for human pressure was established considering trends of tourists and locals at municipality scale;

(iv) In a fourth phase, the Human Sensitivity Index (HSI) (Figure 3) was determined and places investigated were categorized into three sensitive groups, on a 0–1 range of values, according to the same standard previously established for the sensitivity to natural processes (Figure 2).

**Table 3.** Human Impact Index (HI) parameters and Correction Factors used for HSI assessment. * Values used in Mooser et al. [38]. ** Values used for this parameter were slightly modified from the original method [37]. *** New parameter considered for this study.

| Indexes and CF | | | Parameter | Null/Very Low (1) | Low (2) | Medium (3) | High (4) | Very High (5) |
|---|---|---|---|---|---|---|---|---|
| Human Sensitivity Index | Human Impact Index | Visitor pressure | Access difficulty (min) | >45 or only accessible by sea | 25–45 | 10–25 | 5–10 | <5 |
| | | | Protected Area Management Category | Ia & Ib | II & III | IV, V & VI | Only local designation | No |
| | | | Tourism Intensity Rate and Population density * — TIR: tourist beds per 1000 inhabitants ** | <150 | 150–300 | 300–600 | 600–1000 | >1000 |
| | | | PD: persons per km² | <70 | 70–150 | 150–300 | 300–700 | >700 |
| | | | Beach typology ** | Remote | | Rural | | Village or Resort |
| | Correction Factor | | Evolution of the number of beds in tourist establishments (%) ** | Decrease | Minor increase | Increase 15–50% | 50—100% | >100% |
| | | | Evolution of the number of inhabitant (%) *** | Major decrease >25% | Decrease 5–25% | Stable ±5% | Increase 5–25% | Major increase >25% |

Finally, the combination of scores previously calculated for Natural and Human Sensitivity Indexes (NSI and HSI) allowed a Total Sensitivity Index (TSI) to be obtained enabling sites to be classified within corresponding sensitive groups (Figures 2 and 3). Equations employed for the assessment of the indexes, namely EI, NSI, HI, HSI, and TSI and Correction Factors (natural and human) are presented in Appendix A Table A1, and a detailed description of concepts and parameters used can be found in Mooser et al. [37].

## 4. Results and Discussion

### 4.1. Coastal Scenic Beauty (CSES Method)

In total, 16 sites respectively located in Burgas (9), Varna (3) and Dobrich (4) provinces were field-tested during June (2021). The Evaluation Index scores (D) and site scenic characteristics (relating physical and human parameters) are presented in Figure 4 and Table 4. Most sites showed very high values of "D": seven belonged to Class I, corresponding to extremely attractive scenic sites (D ≥ 0.85); eight to Class II (0.65 ≤ D < 0.85); and a single one to Class III (0.40 ≤ D < 0.65), i.e., Vaya PB (Figures 2 and 4). It is noteworthy to mention that most Class II sites could be upgraded to Class I just by applying a few judicious measures (further detailed). Regarding the Vaya PB site, it could easily be upgraded to Class II by reducing beach litter amounts.

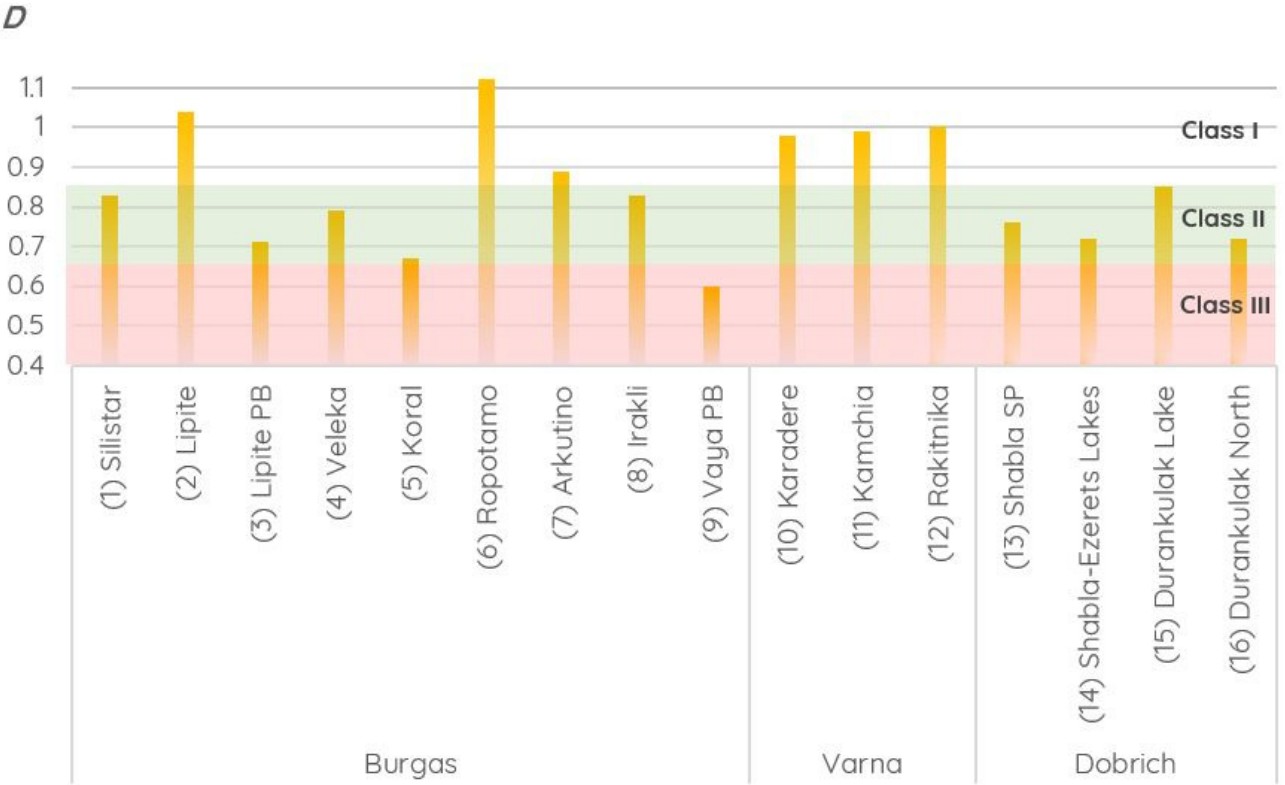

**Figure 4.** Site scores obtained for the Coastal Scenic Beauty Index ("D"), along with the corresponding location map number.

Bulgaria exhibits a splendid variety of scenery from vast plains with many coastal lakes (point 14, Tables 1 and 4; e.g., at Durankulak, Shabla), extensive sand coastlines surrounded by remarkable developed dune systems (point 10; e.g., Ropotamo and Arkutino), shore platform (points 7–9; e.g., Cape Shabla) and high cliff systems (points 1–3; e.g., Rakitnika), to impressive oak pristine forests located in the southern coast of Burgas (point 17; e.g., Silistar, Lipite). Such diversity makes Bulgaria an ideal place to assess coastal scenic beauty. Some sites clearly stand out from the rest with very high scenic values such as Ropotamo (D: 1.12), Lipite (1.04), Rakitnika (1.00), Kamchia (0.99) or Karadere (0.98) (Figure 4). Others showed medium or poor scores for human aspects and were consequently ranged in Class II or III, e.g., Koral (D: 0.67; Class II). General physical and human characteristics were analyzed on the following lines.

4.1.1. Physical Parameters

Excellent scenic values are often linked to the geomorphological setting, e.g., beach type/color and presence of developed dune systems, and the varied coastal physiography. Three basic physiographic systems run from east to west, splitting the whole country into three different regions, including the Danubian plain in the north, undulating and mountain plateaus in the southern part with a transitional area between them. Thus, good scores for "Skyline Landform" (point 12; Tables 1 and 4) were obtained for the Southern Burgas province, i.e., Silistar, Veleka, Arkutino and Ropotamo, whilst the Dobrich province (in the northern part of the coast) was characterized by low values related to flat landforms. Coastal relief along the southern coast of Varna also favored high ratings for "Cliff" parameters at Karadere, Vaya PB and Rakitnika (Figure 5A).

**Table 4.** Site scores obtained from CSES parameters: physical (1–18) and human aspects (19–26).

| Parameter | | 1. Silistar (0.83) | 2. Lipite (1.04) | 3. Lipite PB (0.71) | 4. Veleka (0.79) | 5. Koral (0.67) | 6. Ropotamo (1.12) | 7. Arkutino (0.89) | 8. Irakli (0.83) | 9. Vaya PB (0.60) | 10. Karadere (0.98) | 11. Kamchia (0.99) | 12. Rakit-nika (1.00) | 13. Cap Shabla SP (0.79) | 14. Shabla–Ezerets Lakes (0.75) | 15. Du-rankulak Lake (0.85) | 16. Du-rankulak North (0.72) |
|---|---|---|---|---|---|---|---|---|---|---|---|---|---|---|---|---|---|
| 1–3 Cliff | Height | 1 | 1 | 2 | 1 | 1 | 1 | 1 | 1 | 3 | 2 | 1 | 3 | 1 | 1 | 1 | 2 |
| | Slope | 1 | 1 | 4 | 1 | 1 | 1 | 1 | 1 | 4 | 3 | 1 | 4 | 1 | 1 | 1 | 3 |
| | Features | 1 | 1 | 4 | 1 | 1 | 1 | 1 | 1 | 3 | 3 | 1 | 3 | 1 | 1 | 1 | 3 |
| 4–6 Beach face | Type | 5 | 5 | 4 | 5 | 5 | 5 | 5 | 5 | 5 | 5 | 5 | 5 | 1 | 5 | 5 | 4 |
| | Width | 5 | 4 | 2 | 2 | 2 | 3 | 3 | 4 | 2 | 4 | 4 | 3 | 1 | 3 | 3 | 3 |
| | Color | 4 | 4 | 4 | 4 | 4 | 3 | 4 | 4 | 4 | 5 | 5 | 5 | 1 | 4 | 4 | 4 |
| 7–9 Rocky shore | Slope | 1 | 1 | 3 | 1 | 1 | 1 | 1 | 1 | 1 | 1 | 1 | 1 | 5 | 1 | 1 | 1 |
| | Extent | 1 | 1 | 5 | 1 | 1 | 1 | 1 | 1 | 1 | 1 | 1 | 1 | 5 | 1 | 1 | 1 |
| | Rough | 1 | 1 | 1 | 1 | 1 | 1 | 1 | 1 | 1 | 1 | 1 | 1 | 2 | 1 | 1 | 1 |
| 10. Dunes | | 3 | 4 | 1 | 3 | 5 | 5 | 5 | 1 | 1 | 1 | 5 | 3 | 1 | 4 | 4 | 3 |
| 11. Valley | | 5 | 1 | 1 | 5 | 3 | 5 | 1 | 1 | 1 | 3 | 1 | 1 | 1 | 3 | 1 | 1 |
| 12. Skyline landform | | 3 | 1 | 1 | 3 | 3 | 4 | 4 | 3 | 1 | 3 | 3 | 1 | 1 | 2 | 2 | 1 |
| 13. Tides | | 5 | 5 | 5 | 5 | 5 | 5 | 5 | 5 | 5 | 5 | 5 | 5 | 5 | 5 | 5 | 5 |
| 14. Landscape features | | 3 | 4 | 3 | 4 | 3 | 3 | 4 | 1 | 3 | 2 | 1 | 3 | 3 | 3 | 3 | 2 |
| 15. Vistas | | 3 | 3 | 2 | 4 | 4 | 4 | 4 | 4 | 3 | 4 | 5 | 4 | 4 | 5 | 5 | 4 |
| 16. Water color | | 4 | 4 | 4 | 4 | 4 | 4 | 4 | 5 | 4 | 5 | 5 | 4 | 4 | 5 | 5 | 5 |
| 17. Vegetation cover | | 5 | 5 | 4 | 4 | 5 | 4 | 3 | 5 | 3 | 4 | 5 | 5 | 1 | 3 | 3 | 5 |
| 18. Vegetation debris | | 3 | 3 | 3 | 3 | 2 | 3 | 3 | 1 | 2 | 4 | 4 | 4 | 5 | 3 | 3 | 3 |
| 19. Noise disturbance | | 5 | 5 | 5 | 5 | 5 | 5 | 5 | 5 | 5 | 5 | 5 | 5 | 5 | 5 | 5 | 5 |
| 20. Litter | | 4 | 4 | 3 | 4 | 3 | 4 | 4 | 4 | 3 | 4 | 4 | 4 | 5 | 3 | 4 | 3 |
| 21. Sewage evidence | | 5 | 5 | 5 | 5 | 5 | 5 | 5 | 5 | 5 | 5 | 5 | 5 | 5 | 5 | 5 | 5 |
| 22. NB environment | | 5 | 5 | 5 | 5 | 5 | 5 | 5 | 5 | 5 | 5 | 5 | 5 | 5 | 5 | 5 | 5 |
| 23. Built environment | | 5 | 5 | 5 | 5 | 4 | 5 | 5 | 5 | 5 | 5 | 5 | 5 | 5 | 5 | 5 | 5 |
| 24. Access type | | 5 | 5 | 5 | 4 | 5 | 5 | 5 | 5 | 5 | 4 | 4 | 4 | 4 | 5 | 4 | 4 |
| 25. Skyline | | 5 | 5 | 5 | 3 | 3 | 4 | 3 | 4 | 5 | 5 | 4 | 4 | 4 | 4 | 4 | 5 |
| 26. Utilities | | 1 | 5 | 5 | 2 | 4 | 5 | 5 | 4 | 5 | 5 | 5 | 5 | 5 | 5 | 5 | 5 |

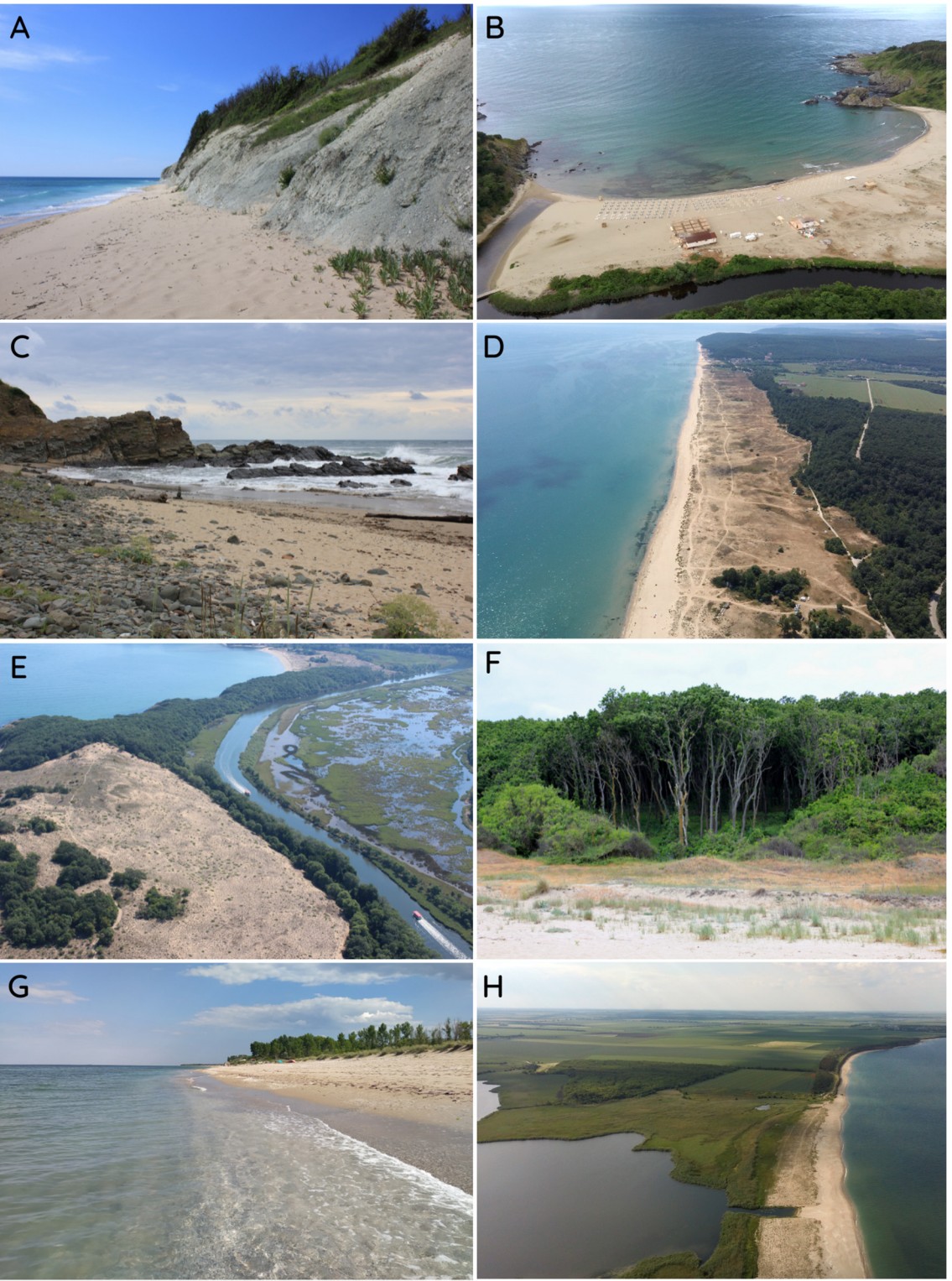

**Figure 5.** Scenic diversity of the Bulgarian coastline: clay cliffs of Karadere, Varna (**A**); river mouth at Silistar, Burgas (**B**); nearshore platform of Lipite PB, Burgas (**C**); Kamchia, the longest Bulgarian beach, Varna (**D**); dune system of Arkutino bordering the famous Ropotamo river, Burgas (**E**); Oak trees reaching the backshore in Lipite, Burgas (**F**); crystal water (**G**) and brackish lake of Sabla–Ezerets, Dobrich (**H**).

Top grades for "Vegetation cover" were particularly observed along the Burgas coastline. The Strandzha Nature Park offers a unique opportunity to see an extremely extensive

oak forest very close to the beach that has been in existence since the end of the Tertiary period (2 million years ago) and is the only example of its kind in Europe. Considered as a "Tertiary living museum", Strandzha is one of the most relevant protected areas in the whole continent in terms of biodiversity (in all biological groups) [58]. The mouths of the Silistar and Veleka rivers are also considered as the most picturesque geotopes on the Bulgarian Black Sea coast [58], and this is reflected by top attribute rates for "Valley" (Figure 5B). The impressive sand spit (around 500 m in length) formed at the Veleka River's mouth was classified under "Special features". At places, rock sectors and headlands give rise to pocket beaches, i.e., at Lipite PB (Figure 5C).

Bulgaria also manifests a wide variety of dune systems. Their distribution commonly depends on the existence of strong onshore winds, coastline orientation, mineralogy and sediment grain size composition. The northern and southern dune systems are situated within the coastal sector of Kamchia and composed of gray dunes with wet dune slacks and forested dunes [58] (Figure 5D). Large dune systems are visible in the northern part of the country, e.g., at Durankulak and Shabla, but the most numerous systems are mainly located along the southern coast, e.g., at Ropotamo, Lipite or Arkutino (Figure 5E). The latter system reaches a maximum height of 50 m, as a result of abundant sediment supplies moving landward under the prevailing NE winds. In Lipite, the striking oaks reach the backshore (Figure 5F). North to Cape Shabla, fore-dunes, dune ridges and fixed stable dunes can be observed and reach 4 m in height at Shabla–Ezerets Lakes. A very detailed description of the Bulgarian dune systems can be found in Stancheva [59] and Stancheva et al. [60].

Most sites showed good scores for "Beach", since the coastline is predominantly composed of fine/medium-grained sand. In the northernmost area occur sand beaches consisting of organic medium-sized sands with high (93%) carbonate contents [61] because of the large mussel fields found in the nearshore. Beaches here have a low heavy mineral content, reflected by a top rating for "Beach color", e.g., Rakitnika or Karadere (white/gold color; Tables 1 and 4). Within the central coast, sand beach sediment input is basically from landslides and small rivers. Generally, beach sands are coarse to medium grain-sized, with low carbonate content consisting predominantly of quartz. At the southernmost section, beaches are composed of medium and fine grain sized magnetite–titanite sands, with a high content of heavy minerals (up to 75%) due to volcanic rocks [62], e.g., Ropotamo (dark tan color; rated 3, Table 4). Just one place had a large rock shore platform, i.e., Cape Shabla, and clear or crystal water was observed for "Water color and clarity" (point 16, Table 1) at almost all investigated sites (Figure 5G).

Finally, close to the border with Romania, coastal lakes were observed at Durankulak and Shabla–Ezerets sites (Dobrich) (Figure 5H). These brackish lakes are surrounded by fields, shrubs and separated from the Black Sea by narrow sand bars. In the case of Shabla, a connection between two lakes was made with a thin artificial canal. Situated on the Via Pontica, the second largest bird migratory route in Europe, they constitute an essential stopover or wintering refuge for many bird species and host a large variety of endemic plants, being both Ramsar sites. From a scenic perspective, these elements were reflected by good scores at "Coastal landscape features" (point 14, Table 1).

### 4.1.2. Anthropogenic Parameters

Scores obtained for human parameters (CSES method) [39–41] are presented in Figure 6. Located in remote (13) and rural areas (3), sites frequently showed top ratings for "Noise", "Sewage", "Built" and "Non-Built environment". However, some significant variances were observed for "Litter", "Skyline", "Utilities" and, to a lesser extent, "Access type" (Figure 6). The discussion is focused on these parameter scores and on the proposal of judicious interventions for their improvement.

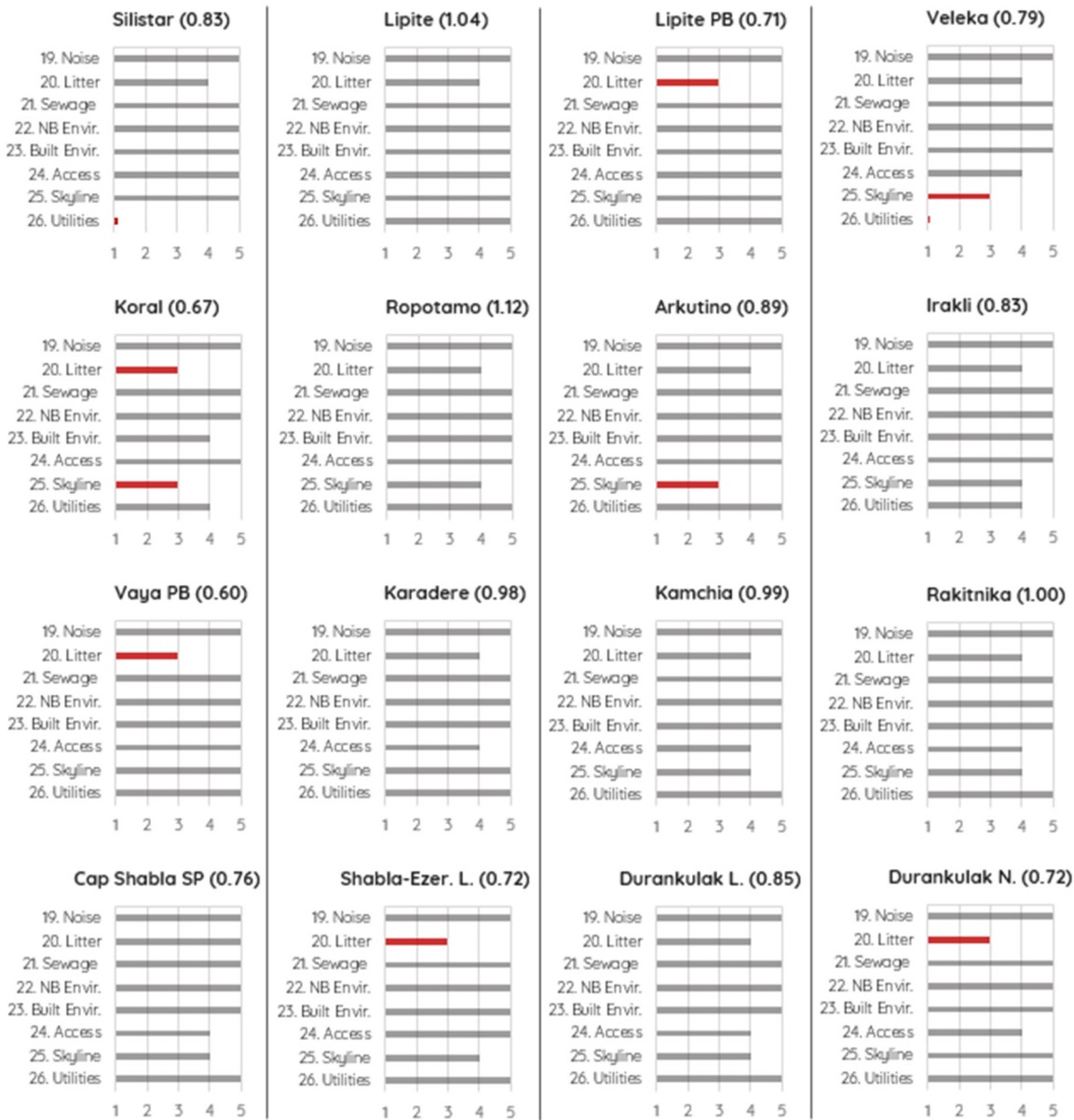

**Figure 6.** Scores obtained for human parameters by the CSES method (values ≤3 are in red).

4.1.3. Analysis and Suggestion Measures

(a)    General analysis of Classes I and II

All investigated sites (apart from Koral) are located in protected areas belonging to different and complementary designations types, at regional, national (e.g., Nature Parks), European (e.g., SCI and SPA; Natura 2000) and/or international levels (e.g., Ramsar). The Strandzha Nature Park (Burgas) located along the southern coast, was the area most represented in this study with three sites, i.e., Silistar, Lipite and Lipite PB. Burgas was also the province that contained most sites (nine; Figure 1).

As stated previously, seven sites (out of 16) corresponded to Class I, eight to Class II and only one to Class III. Below, two distinctive examples of Class I and Class II, respectively Ropotamo and Veleka, were selected to characterize both classes. Their ratings, membership degree curves and weighted averages can give an immediate visual state of the scores obtained by relating the physical and anthropogenic parameters (Figures 7 and 8). Examples of Class III are not presented, because only one site was found, i.e., Vaya PB.

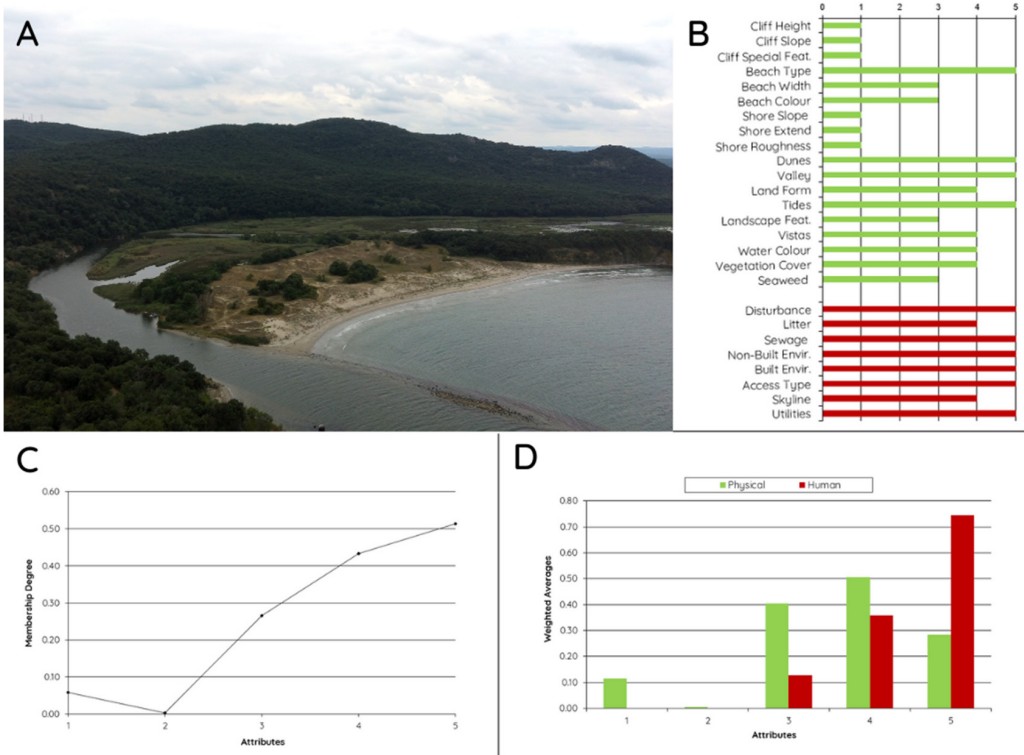

**Figure 7.** Ropotamo beach (**A**) and corresponding CSES ratings (**B**); membership degree vs. attributes (**C**) and weighted averages vs. attributes (**D**).

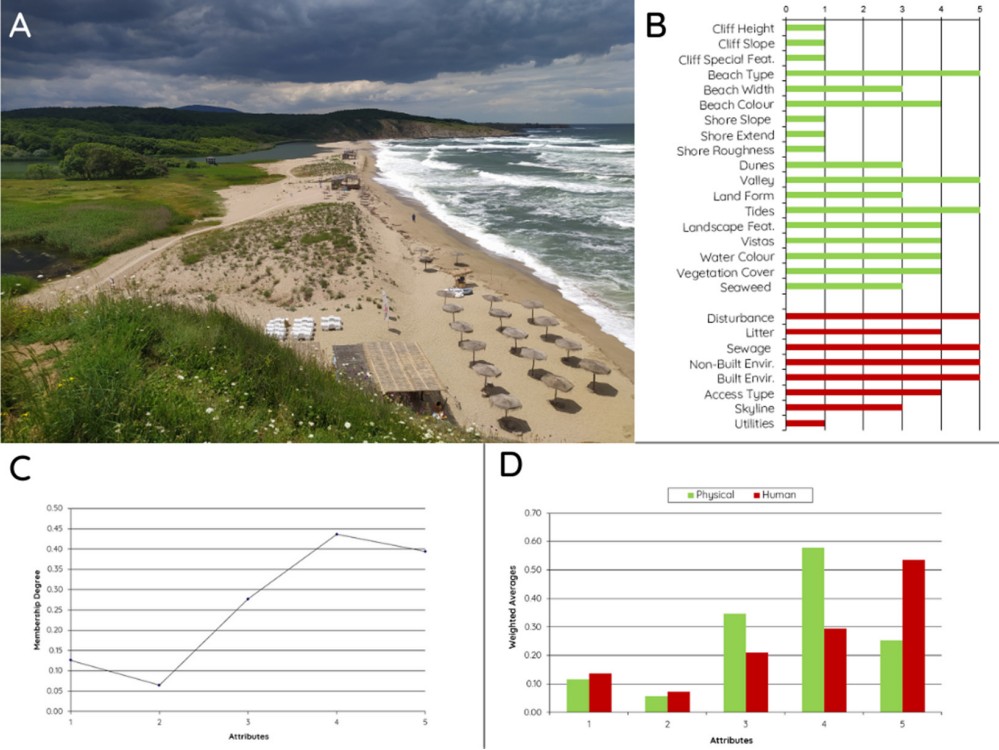

**Figure 8.** Veleka beach (**A**) and corresponding CSES ratings (**B**); membership degree vs. attributes (**C**) and weighted averages vs. attributes (**D**).

**Class I** sites are extremely attractive with outstanding features represented by physical and anthropogenic parameters (D ≥ 0.85). Located in a Strict Nature Reserve, Ropotamo

(D: 1.12) is the perfect illustration of a wild and remote area with restricted access, where human impact is almost non-existent (Figure 7A–C). Testing this place involved climbing the Arkutino' dune system, the crossing of a dense riparian forest of oak, ash, elm, etc., and the skirting of the eponymous river (entailing a 75-min walk). All human scores showed top grades, except for "Litter" and "Skyline", both rated 4. With regard to physical aspects, very good scores were observed for "Beach type" (sand), "Dunes", "Valley" (the river mouth is around 30 m in width), "Landform", "Vistas", "Water color" and "Vegetation cover" (Figure 7A,B). General physical and human ratings (Figure 7C,D) led to a very high "D" value (1.12). A few sites belonged to State Game Husbandries, e.g., Arkutino, a very common protection feature in Bulgaria that offers less protection than the previously cited features.

**Class II** refers to attractive natural sites with a low intrusion of human impact with "D" values 0.85 > D ≥ 0.65. These sites frequently rated lower than Class I due to a lower scoring of the physical parameters, e.g., Irakli (low scores for "Special Features", "Dunes", "Vegetation debris"), or because of the influence of human activities, e.g., Veleka. Chosen as an example in Figure 8, the latter showed top ratings for "Beach" (type and color), "Valley", "Coastal landscape features" (among them an impressive sand spit), "Vistas" and "Vegetation cover". However, low scores linked to the skyline quality and to very intrusive "Utilities" (Figure 8A–D) significantly downgraded the environmental richness. The same happened at Silistar, Koral, Shabla–Ezerets Lakes or Durankulak North, where human related activities critically lowered their natural attractiveness, downgrading them to Class II.

Finally, it is interesting to highlight which human impacts are observed in Class I and Class II (Figure 9A,B). Results clearly reflect how litter adversely affects the scenic beauty of investigated sites. Likewise, critical attention should be paid to any improvement in "Utilities", since low scores observed of this parameter enables a potential upgrade of sites with high physical values (Figure 9C) to Class I, e.g., Veleka or Silistar.

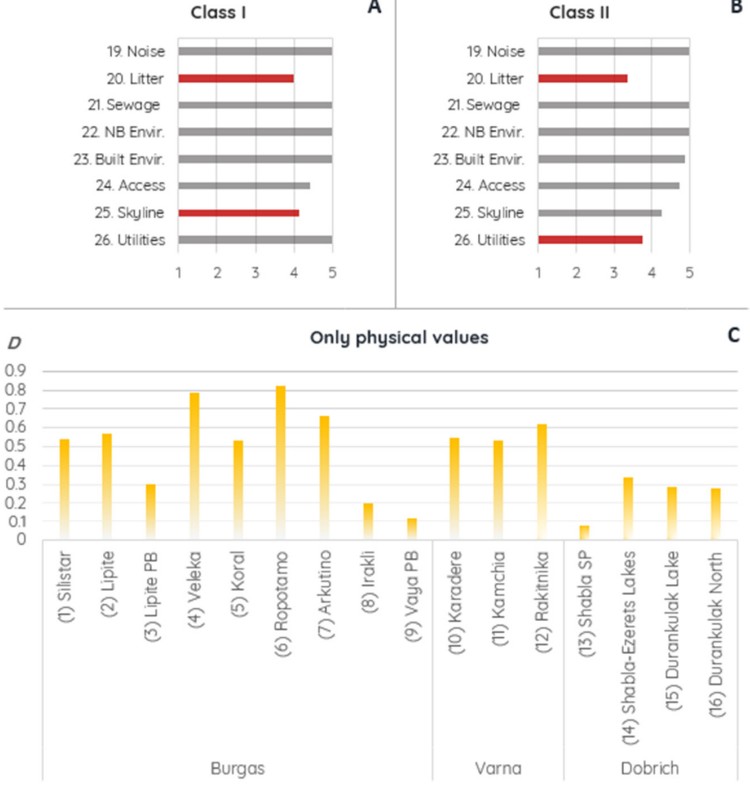

**Figure 9.** Human scenic impact observed in Class I and Class II (lowest scores in red) (**A**,**B**); sites' scenic beauty without considering human parameters (**C**).

Parameter values and suggestions for scenic enhancement of physical components, e.g., dune strengthening, rehabilitation and restoration, or beach nourishment. This is why emphasis should be placed on coastal managers to work out ways of upgrading anthropogenic parameter scores. Turning the clock back is not feasible, and certain scenic impacts, such as skyline quality, are virtually irreversible. However, several small and judicious interventions with high effectiveness may be achieved to upgrade any sites' attractiveness within determined areas, e.g., by reducing litter or utilities. Scores of human parameters were analyzed, and suggestions to managers are presented.

(1) **Noise disturbance** was non-existent at all investigated sites during field-work observations. However, it should be noted that scores obtained at Veleka (D: 0.79) and Silistar (0.83) could substantially vary during the peak tourist season. Indeed, both sites tend to considerably increase their number of visitors, as they are easily accessible by a <10-min walk and allow for the presence of several beach bars, together with a large number of sunbeds. Many tourists could decrease both sites' attractiveness to 0.73 and 0.79, respectively.

(2) **Litter,** linked to discharged items proceeding from different sources, was mainly characterized by single accumulations (rated 3; Table 1 and Figures 6 and 9) and a few scattered items (rated 4). Litter items observed along the study area were mainly stranded by sea currents and rivers, and they were usually composed of plastic items (bottles, bags and cups), glass drinks bottles, fast-food packaging, cans, foamed polystyrene and ship waste (e.g., shipping rope) (Figure 10A–D). Their presence critically lowered a site's rating. As observed in Andalusia or the Balearic Islands, the absence of periodic cleaning operations is probably due to the difficult access for cleanup machines [38,53]. However, the fact that most items have been lying on the beach for several years constitutes stark evidence of the low interest of competent authorities and managers. Litter has a very large impact on coastal tourism and recreation [63]. As a way of illustrating the relationship between beach litter and its scenic impact, if the current litter rating (3) observed at Lipite PB (D:0.71), Koral (0.67), Vaya PB (0.60), Shabla–Ezerets Lakes (0.75) and Durankulak North (0.72) is upgraded +1, the "D" value of these sites will respectively jump to 0.86, 0.80, 0.74, 0.89 and 0.88. These interventions would upgrade Lipite PB and the two latter to Class I, as well as Vaya PB to Class II. Only at Cap Shabla SP was litter virtually absent.

(3) **Sewage** was not evident at the sites investigated. Its presence is frequently visible in urban or village beach typologies [57], but hardly ever in remote areas.

(4) **Non-Built Environment** is the environment as perceived minus its buildings. In the case of Dobrich district, a very agricultural region located in the Northern Bulgaria, fields were relatively close but not visible from the beach, e.g., Durankulak North. All sites gave top scores (5) (Figure 6).

(5) **Built Environment** refers to surrounding anthropogenic structures, buildings, etc. Sites obtained top values (5), as they were located in the natural environment, except at Koral, where several bungalows are found around the beach (in its northern sector); this was characterized as "sensitive tourism" (rated 4).

(6) **Access type** usually showed good scores (≥4). Site lower scoring was principally due to four-wheel-drive vehicles that illegally crossed dune systems to get as close as possible to the beach to carry out recreational or fishing activities, among others, i.e., Kamchia, Rakitnika, Durankulak Lakes and Durankulak North. Beyond the scenic impact, this bad practice raises concerns about dunes and beach users' protection; this critical point is further discussed (cf. sensitivity section). Another curious case is Karadere, where the presence of motor homes and caravans were noticed in the backbeach, due to a lack of camping restrictions (Figure 10E). At Veleka, "Access type" was ticked attribute 4, since an unpaved road was visible from the beach.

(7) **Skyline** alludes to buildings' silhouettes not in harmony with the environment. Top grades are frequently related to sites having restricted views, e.g., Vaya and Lipite pocket beaches. Large coastal sectors, as Durankulak Lake, Shabla–Ezerets Lakes

or Irakli, located close to sensitively designed human settlements (without high buildings), obtained good scores (4). Kamchia rating (4) was linked to the presence of a 200 m pier emplaced in Shkorpilovtsi village. The worst scores (3) were noticed at Veleka, Arkutino and Koral (Figure 10F,G). The first two are located near the borders of protected areas, while the latter is out of any protected area. In the case of Veleka (D: 0.79; Class II), if the municipality would have not allowed the construction of a few elevated buildings (4 or 5 floors) near the beach rather than traditional houses (with low heights), the "D" value would reach 0.93 (Class I). At Arkutino (D: 0.89), a polemic unfinished resort complex whose construction was abandoned in the late 1980s remains relatively close to the beach (northern sector); without it, Arkutino would be one of the top Bulgarian scenic sites (D: 1.03; without skyline impact).

(8) **Utilities** is the parameter that covers a large variety of human items, e.g., power lines, lighting, pipelines, seawalls, revetments or temporary leisure facilities, amongst others. Most sites had good scores (4 or 5), except for Silistar and Veleka (rated 1; Figure 6). In both cases, their scenic impact was associated with intrusive structures devoted to seasonal use, i.e., several beach bars, beach umbrellas, first-aid stations and hundreds of sun beds (Figure 10G). This is the perfect illustration of one of the major issues that coastal managers must resolve in "3S" destinations where conflicts arise between scenic preservation and short-term benefits. Such a dilemma was also observed in Andalusia or Balearic Islands, among many other destinations [38,53]. If the administration of Strandzha Nature Park (both sites are in the Strandzha Nature Park) was not so permissive in relation to leisure facilities (but allowing first aid stations), the attractiveness of Silistar (0.83) and Veleka (0.79) could respectively jump to 0.97 and 0.92, upgrading both sites to Class I. At numerous places, lifeguard stations are certainly indispensable because of rip currents, but beach bars and other utilities (sun beds, beach umbrellas, beach kiosk, etc.) should be reduced and/or moved away from the beach, preserving the essence of natural sceneries. Beaches have to be managed according to their typologies and not as a whole. From a management approach, it is not rational that some remotes sites provide the same services as carried out on resort or urban beaches. Finally, at Koral, the presence of two old fallen lifeguard towers gave an attribute rating of 4 (Figure 10H); at Rakitnika, gas pipelines (linked to the offshore Galata Platform) were not considered, as they were not visible from the beach (covered by sand).

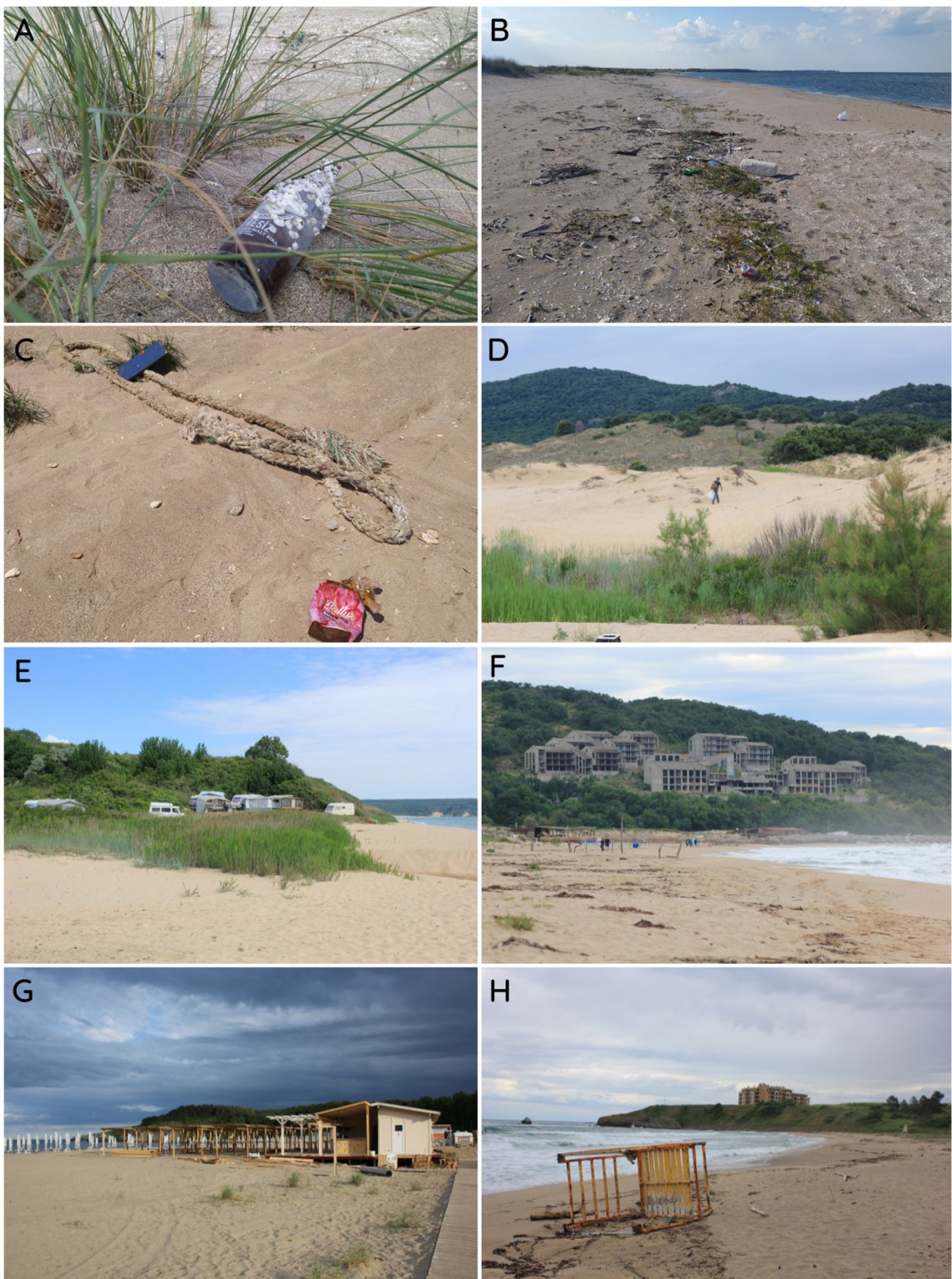

**Figure 10.** Human impact on scenery and examples of "Litter", "Access", "Skyline" and "Utilities": glass bottle (from Turkish producer) stranded by sea currents (**A**); cans, plastic bottles and foam polystyrene dropped off at the shoreline Shabla–Ezerets Lakes, Dobrich (**B**); discarded ship rope and can of adhesive paint probably dropped from a Turkish ship, Dobrich (**C**); inspiring good practices observed at Arkutino, Burgas (**D**); cars and motorhomes setting up camp near the beach of Karadere, Varna (**E**); detailed view of skyline impact caused by unfinished constructions at Arkutino (**F**); seasonal utilities at Silistar, Burgas (**G**); abandoned lifeguard tower and building construction in the background at Koral, Burgas (**H**).

### 4.2. Coastal Scenic Sensitivity (CSSI Method)

As mentioned above, scenic sensitivity to natural processes and human pressure was quantified by using the CSSI method [37,38]. General site characteristics and scores obtained for the Natural Sensitivity Index (NSI), Human Sensitivity Index (HSI) and Total Sensitivity Index (TSI) are presented in Table 5. The results can be useful to prevent and limit future environmental degradation linked to natural processes, in a climate-change context, and human activities in coastal areas of great scenic values, as well as to suggest measures to improve their resilience.

**Table 5.** Main site characteristics: provinces (Pr.), typologies, beach length, protected areas types (National, Natura 2000 and Ramsar) with corresponding IUCN categories, Scenic Sensitivity Indexes (CSSIs) and "D" values (CSES). * Acronyms: pocket beach (PB); shore platform (SP). ** Acronyms: Nature Park (NP); Site of Community Importance (SCI); Special Protection Area for Birds (SPA).

| Site * | Pr. | Typology | Length (m) | Protected Areas and IUCN Category ** | NSI | HSI | TSI | D |
|---|---|---|---|---|---|---|---|---|
| 1.Silistar | | Remote | 462 | Strandzha NP (V) SPA and SCI | 0.84 | 0.59 | 0.72 | 0.83 |
| 2. Lipite | | Remote | 380 | Strandzha NP (V) Silistar Protected Area (VI) SPA and SCI | 0.77 | 0.47 | 0.62 | 1.04 |
| 3. Lipite PB | | Remote | 62 | | 0.59 | 0.53 | 0.56 | 0.71 |
| 4. Veleka | | Rural | 838 | Strandzha NP (V) Veleka Protected Area (VI) SPA and SCI | 0.81 | 0.58 | 0.70 | 0.79 |
| 5. Koral | Burgas | Rural | 765 | None terrestrial (only SCI marine) | 0.58 | 0.67 | 0.63 | 0.67 |
| 6. Ropotamo | | Remote | 547 | Ropotamo Strict Nature Reserve (Ia) SCI and SPA Ramsar Estuary of the Ropotamo River Ropotamo State Game Husbandries | 0.64 | 0 | 0.32 | 1.12 |
| 7. Arkutino | | Remote | 1005 | Ropotamo State Game Husbandries SCI and SPA | 0.66 | 0.63 | 0.65 | 0.89 |
| 8. Irakli | | Remote | 3829 | Irakli Protected Site (VI; southern sector) Nessebar State Game Husbandries SCI and SPA | 0.66 | 0.69 | 0.68 | 0.83 |
| 9. Vaya PB | | Remote | 328 | Nessebar State Game Husbandries SCI and SPA | 0.89 | 0.63 | 0.76 | 0.60 |
| 10. Karadere | | Remote | 3771 | SCI and SPA Natural Monuments (III, southern sector) | 0.92 | 0.63 | 0.78 | 0.98 |
| 11. Kamchia | Varna | Remote | 6140 | SPA and SCI Strict Nature Reserve (Ia, Kamchia River outlet) Protected Site (VI, Kamchia River outlet) | 0.58 | 0.56 | 0.57 | 0.99 |
| 12. Rakitnika | | Remote | 1349 | Rakitnika Protected Site (VI; northern sector) Liman Protected Site (VI; southern sector) SCI (northern sector) SPA | 0.66 | 0.56 | 0.61 | 1.00 |
| 13. Cape Shabla SP | | Rural | 1738 | Balchik State Game Husbandries SPA | 0 | 0.59 | 0.30 | 0.76 |
| 14. Shabla–Ezerets Lakes | Dobrich | Remote | 2820 | Shablensko Ezero Protected Site (VI) SCI and SPA Ramsar Lake Shabla Balchik State Game Husbandries | 0.50 | 0.38 | 0.44 | 0.72 |
| 15. Durankulak Lake | | Remote | 6770 | Ezero Durankulak Protected Site (VI) SCI and SPA Ramsar Lake Durankulak Balchik State Game Husbandries | 0.53 | 0.38 | 0.46 | 0.85 |
| 16. Durankulak North | | Remote | 1915 | Balchik State Game Husbandries SCI and SPA | 0.59 | 0.50 | 0.55 | 0.72 |

### 4.2.1. Sensitivity to Natural Processes

This index aims to determine the intrinsic scenic sensitivity of most attractive coastal sectors to erosion and/or flooding processes by considering their scenic characteristics, the level of potential stress caused by forcing variables and, finally, the predictions of Relative Sea-Level Rise (RSLR) and Storm Surge (SS) by 2100. The location, length and values obtained for NSI are presented in Figure 11. The sectors considered are characterized by

homogeneous scenic values relating the physical and human aspects. For example, in the Kamchia study case, the most extensive Bulgarian beach (12.4 km), "only" 6.1 km of length was considered in this study (from the eponymous river mouth to the beginning of Shkorpilovski village), since its northern and southern sectors are surrounded by resorts and settlements linked to nearby villages, e.g., houses and a pier impacting on scenic beauty.

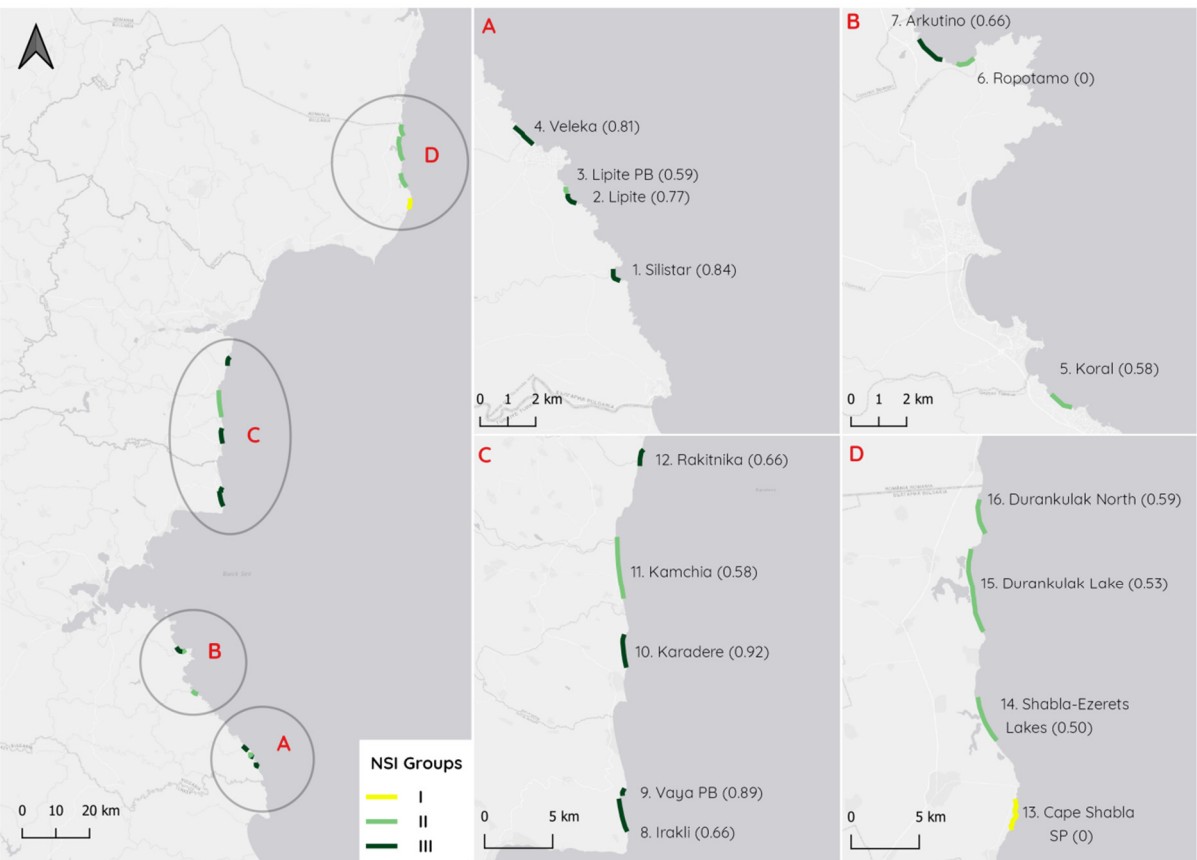

**Figure 11.** Investigated sectors, global view on the left map and zooms of (**A**–**D**) zones, with NSI values and the corresponding sensitive group.

First, the sites were divided into three categories according to their physical characteristics and scores previously obtained via the CSES method (Figure 3). Only Cape Shabla SP, characterized by a large shore platform, was considered as not sensitive and therefore included in Category I and not further investigated (Figure 11). A few sites that showed "Beach face" but no "Dunes" (≤2; CSES) were ranked in Category II (3), i.e., Lipite PB, Vaya PB and Karadere, whilst most of them belonged to Category III (12), as they presented good scores for both parameters (Tables 2 and 6). Next, the following parameters related to "Beach face" and "Dunes" were assessed, enabling an Erodibility Index (EI) for locations classified in Categories II and III (Table 6). All parameters were rated on a five-point attribute scale, with 1 indicating a great contribution of a specific key variable to site resilience and 5 indicating a low contribution/a high sensitivity.

**Table 6.** Site scores for NSI parameters. * Category I site: No further investigation is required.

| Site | Province | Category | Dry Beach | Sediment | RS Width | RS Location | Dunes Height | Dunes Width | Vegetation Cover | Washovers | EI | Hs | Angle of Approach | Tidal Range | Storm Surge | Sea-Level Rise | NSI | Group |
|---|---|---|---|---|---|---|---|---|---|---|---|---|---|---|---|---|---|---|
| 1. Silistar | Burgas | III | 5 | 5 | 5 | 5 | 4 | 4 | 3 | 4 | 0.90 | 3 | 5 | 5 | 1 | 5 | 0.84 | III |
| 2. Lipite | | III | 5 | 5 | 5 | 5 | 3 | 4 | 2 | 1 | 0.79 | 3 | 5 | 5 | 1 | 5 | 0.77 | III |
| 3. Lipite PB | | II | 5 | 3 | 3 | 1 | | | | | 0.58 | 3 | 3 | 5 | 1 | 5 | 0.59 | II |
| 4. Veleka | | III | 5 | 5 | 5 | 5 | 3 | 4 | 3 | 4 | 0.88 | 3 | 3 | 5 | 1 | 5 | 0.81 | III |
| 5. Koral | | III | 2 | 5 | 5 | 5 | 2 | 2 | 1 | 2 | 0.56 | 3 | 3 | 5 | 1 | 5 | 0.58 | II |
| 6. Ropotamo | | III | 5 | 5 | 5 | 5 | 1 | 1 | 1 | 1 | 0.67 | 3 | 1 | 5 | 1 | 5 | 0.64 | II |
| 7. Arkutino | | III | 5 | 5 | 5 | 5 | 1 | 1 | 1 | 1 | 0.67 | 3 | 3 | 5 | 1 | 5 | 0.66 | III |
| 8. Irakli | | III | 1 | 5 | 5 | 5 | 4 | 4 | 3 | 3 | 0.65 | 3 | 5 | 5 | 1 | 5 | 0.66 | III |
| 9. Vaya PB | | II | 5 | 5 | 5 | 5 | | | | | 1.00 | 3 | 1 | 5 | 1 | 5 | 0.89 | III |
| 10. Karadere | Varna | II | 5 | 5 | 5 | 5 | | | | | 1.00 | 3 | 5 | 5 | 1 | 5 | 0.92 | III |
| 11. Kamchia | | III | 1 | 5 | 5 | 5 | 2 | 2 | 2 | 3 | 0.55 | 3 | 5 | 5 | 1 | 5 | 0.58 | II |
| 12. Rakitnika | | III | 1 | 5 | 5 | 5 | 4 | 4 | 3 | 4 | 0.67 | 3 | 3 | 5 | 1 | 5 | 0.66 | III |
| 13. Shabla–Ezerets SP * | Dobrich | I | | | | | | | | | 0.00 | | | | | | 0.00 | I |
| 14. Shabla–Ezerets Lakes | | III | 1 | 3 | 5 | 5 | 2 | 4 | 2 | 2 | 0.46 | 3 | 3 | 5 | 1 | 5 | 0.50 | II |
| 15. Durankulak Lake | | III | 1 | 3 | 5 | 5 | 3 | 4 | 3 | 2 | 0.50 | 3 | 3 | 5 | 1 | 5 | 0.53 | II |
| 16. Durankulak North | | III | 2 | 3 | 5 | 5 | 4 | 4 | 3 | 2 | 0.58 | 3 | 3 | 5 | 1 | 5 | 0.59 | II |

(1) **Dry beach width as a multiple of the ICZ** was calculated comparing shorelines for the period 1972–2011, using topographic maps (1:5000) and orthophotos (images from 2019 were only available for Dobrich province). Half of the sites were rated 5, since high recorded erosion rates indicated significant beach width loss ("Dry beach", Table 6). Sectors such Silistar, Karadere or Vaya PB lost respectively 8.47 m (i.e., a beach width of 40 m), 8.36 m (21.5 m) and 14.48 m (4 m) during the investigated period. Only five, mainly located in the central–northern part of the country, manifested stability or accretion rates (rated 1), i.e., Irakli, Kamchia, Rakitnika, Shabla–Ezerets Lakes and Durankulak Lake (Table 6). The last two abovementioned sites increased their width by 8.62 m and 6.72 m respectively. Two sites, Koral and Durankulak North, gave a rating of 2, as they presented slight erosion rates coupled with values of "Dry beach" >4 times the ICZ (Table 6).

(2) **Sediment grain size** showed high ratings, since most sites (11) were composed of fine-grained sand (rated 5, Table 6). Four mixed beaches, mainly consisting of sand and, to a lesser extent, pebbles, gravel and/or broken shells, obtained intermediate scores (3) (Table 6). Curious cases were noticed in the northern sectors of Shabla and Durankulak where very impressive accumulations of black shell mussels remained on the beach shoreline (Figure 12A). At these places, reefs constitute the main source of beach material, providing over 90% of sediments [61].

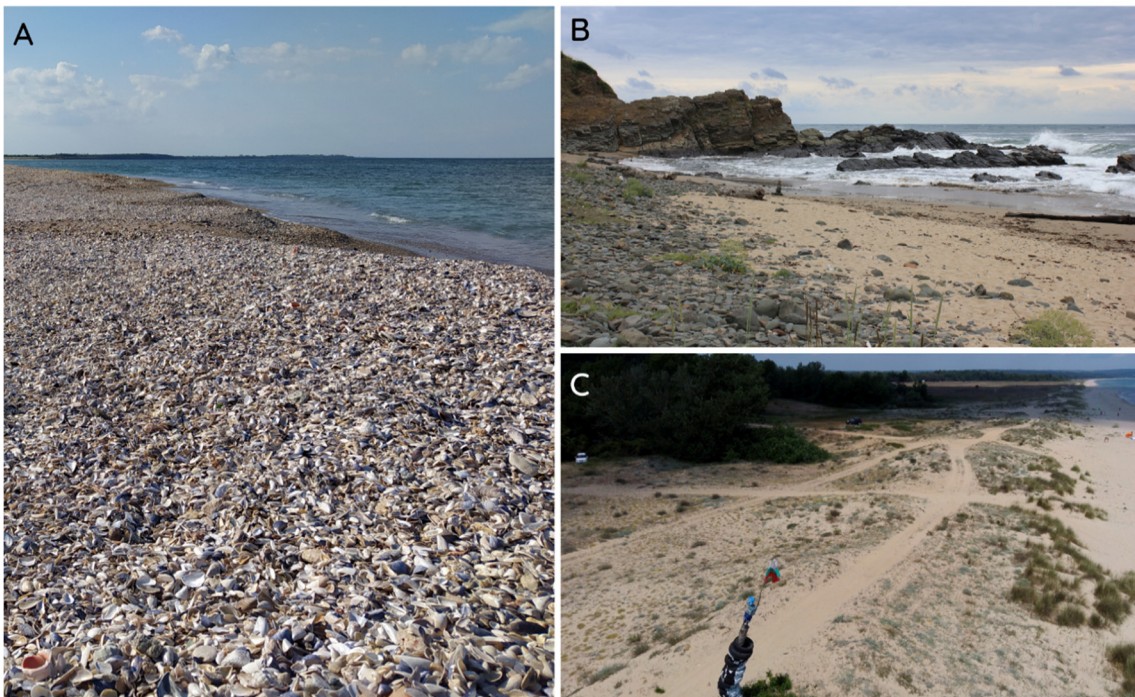

**Figure 12.** Fragmented shells at Shabla–Ezerets Lakes (**A**), mixed-beach and nearshore platform dissipating wave energy at Lipite PB (**B**) and illustration of adverse effects linked to vehicle circulation on the dune system at Kamchia (**C**).

(3) **Rocky shore width and location**—only Lipite PB exhibited a large and emerged nearshore platform dissipating wave energy that provided a natural defense to erosion processes (Figure 12B). This was reflected by good scores for both parameters (Table 6).

(4) **Dune parameters**, including dune height, width, vegetation cover and washovers, were considered for sites belonging to Category III. Very strong and healthy dune systems that are highly resilient to potential stressing events were recorded at Arkutino and Ropotamo, giving the lowest values (1) for each parameter (Table 6). Low grades were also observed at Kamchia and Koral. However, in the case of Kamchia (and the rest of the northern and central part of the country), the illegal use of vehicles and their

associated adverse effects, e.g., fragmentation, loss of vegetation and biodiversity, displacement, compaction, etc., is a very serious issue that beach managers must resolve. For example, at Kamchia and Rakitnika, washover fans, whose formation was favored by this bad practice, broke the dune ridge continuity forming sensitive hot spots to coastal erosion (Figure 12C). At many sites, dune width was also considerably reduced or fragmented by trails parallel to the coast that are mainly used by off-road vehicles, i.e., Kamchia, Shabla–Ezerets Lakes, Durankulak Lake and Durankulak North. To give an instance of effective dune management, a place such as Rakitnika (NSI: 0.66) could improve its general NSI to 0.61 only by reducing washovers <25% and increasing dune width up to 50 m. In the southern part of the country, Veleka's (NSI: 0.81) and Silistar's (NSI: 0.84) high scorings were partially associated with the high level of recreational activities and related impacts forming critical gaps in the dune ridge, leading to a loss of vegetation cover and dune width. By mitigating the cumulative effects of pedestrians and beach bars presence, both sites could respectively decrease their sensitivity to 0.75 and 0.77. The real effectiveness of these measures is relative (and probably undervalued), as it is hard to predict how they could influence/reduce the rates of shoreline erosion in future decades. The lowering of the current scores recorded in the first parameter, "Dry Beach as multiple of the ICZ" (rated 5 for both sites, Table 6), would greatly increase the resilience of such coastal features.

Finally, a Correction Factor (CF) was estimated by taking into account forcing variables and regional predictions of RSLR and SS by the end of the century [37].

(1) **Forcing variables** include "Wave characteristics" and "Tidal range" parameters. The second was characterized by the highest grade (5) (Table 6), since microtidal coasts are most exposed to potential storm events, as they are always near high tide—a large amount of the literature supports this viewpoint [64–66]. Results for "Significant wave height" ($H_s$) and "Angle of wave approach" were extracted from the Bulgarian National Oceanographic Data Centre [67]. Only three virtual buoys, respectively located in Burgas, Shkorpilovski and Varna, were analyzed during the winter period from October 2020 to March 2021—due to the lack of long time series and scarcity of virtual buoys along the study area—leading to the following values: Varna ($H_s$: 1.01 m; 90–95°), Shkorpilovski ($H_s$: 1.14 m; 80–90°) and Burgas ($H_s$: 0.97 m; 75–85°). Given this context, all sites obtained a rating of 3 for $H_s$ (0.75–1.5 m) (Table 6), whilst the "Angle of wave approach" was judged relating to each site location, varying from 1 to 5, e.g., Ropotamo (1; oblique 40°), Silistar (5; parallel 0°) (Table 6). It should be noted that the Shkorpilovski buoy recorded the highest energy event, with waves reaching around 5.70 m in height in March 2021.

(2) **Ongoing changes in RSLR and SS** are due to anthropogenic climate change and other factors, and they present a global challenge to coastal managers. It is acknowledged that the Black Sea and its coastal zones are one of the most sensitive areas in Europe at risk for coastal erosion and saltwater intrusion [68]. For European countries, Mean Sea Level (MSL) is expected to reach around 53 cm and 77 cm, under the Representative Concentration Pathways 4.5 and 8.5 (RCP), while projections for the Black Sea give around 59 cm and 80 cm by 2100 [69]. According to Volkov and Landerer [70], the forcing of sea level in the Black Sea is dominated by the basin freshwater budget and water exchange through the Bosporus Strait, as well as depth-integrated changes in seawater density. Many studies have reported that MSL reaches the highest levels during the May–June period [68,70]. RSLR predictions obtained from the LISCOAT database [71] gave a rating of 5 for all investigated sites (Table 6), with values varying from 0.44 (RCP 4.5) to 0.71 (RCP 8.5). Data gaps in tide gauge stations did not allow for estimates of a reliable trend for potential local subsidence effect [68,72]. The storm surge level, defined as the difference between the pure tide and the total water-level simulations, was estimated by using the Copernicus dataset of "Sea level indicators for the European coast from 1977 to 2100" [73]. Based upon past observational data and future climate projections at any regional scale, predictions around 35 cm (for 2100)

were recorded for the entire Bulgarian coast; this was reflected by a low scoring (1) (Table 6).

Combining the EI and the CF values as specified by Mooser et al. [37], we obtained the following NSI values for the 16 investigated sites (Figures 12 and 13 and Table 6), enabling their classification into one of the three sensitive groups (Figure 3).

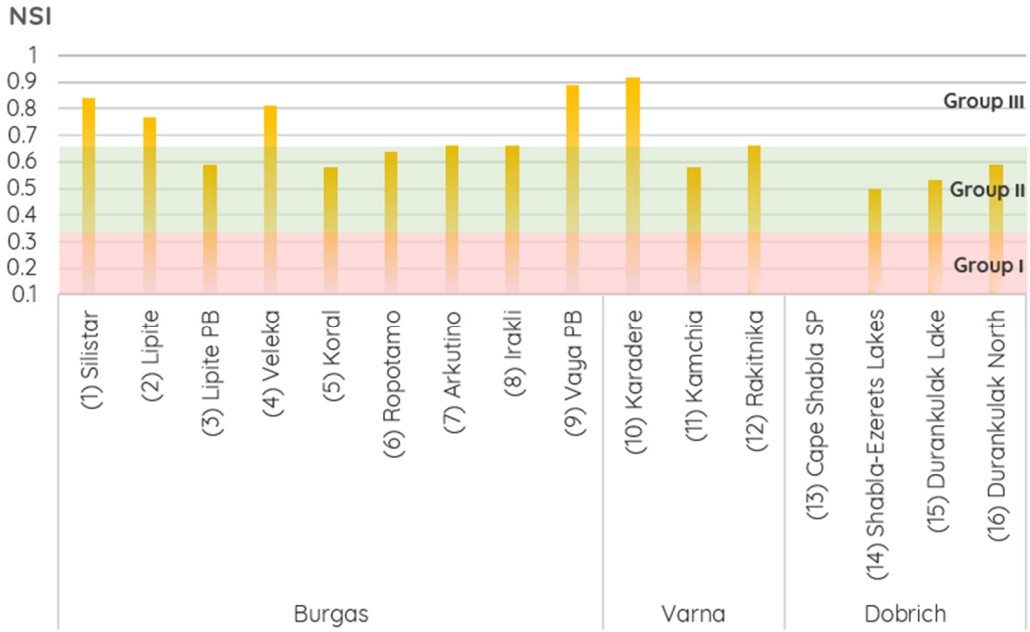

**Figure 13.** NSI values recorded at sites investigated, along with corresponding sensitive group.

4.2.2. Sensitivity to Human Pressure

In a global scenario of coastal unsustainable growth, this section aims to determine a sites scenic sensitivity to visitor pressure and their perception to scenery and human settlements [74], considering, as Correction Factors, local trends at the municipality scale of tourists and locals. The above enabled the calculation of a Human Sensitivity Index (HSI) presented in Figure 14 with their corresponding location and sensitive group.

Sites were firstly included in one of the three scenic categories detailed by Mooser et al. [37], in agreement with their location and ratings previously obtained for human parameters by the CSES method. Only Ropotamo belonged to Category I (Table 7 and Figure 14), as it is the only place located in a very remote area (a walk of around 1-h walking) and under a strong protection category, i.e., Strict Natural Reserve—its sensitivity to human factors was not investigated. Almost all locations were considered as Category II (12), showing low scoring for human impact, mainly associated with "Litter" and "Utilities" (Table 7 and Figure 14). Only three fell into Category III, due to a major scenic impact at "Skyline", "Built Environment" or "Access type", principally linked to their typology, i.e., Veleka, Koral and Cape Shabla SP (Table 7 and Figure 14). The following parameters were evaluated for all investigated sectors (except Ropotamo), allowing for the calculation of a Human Impact Index (HI) (Table 7).

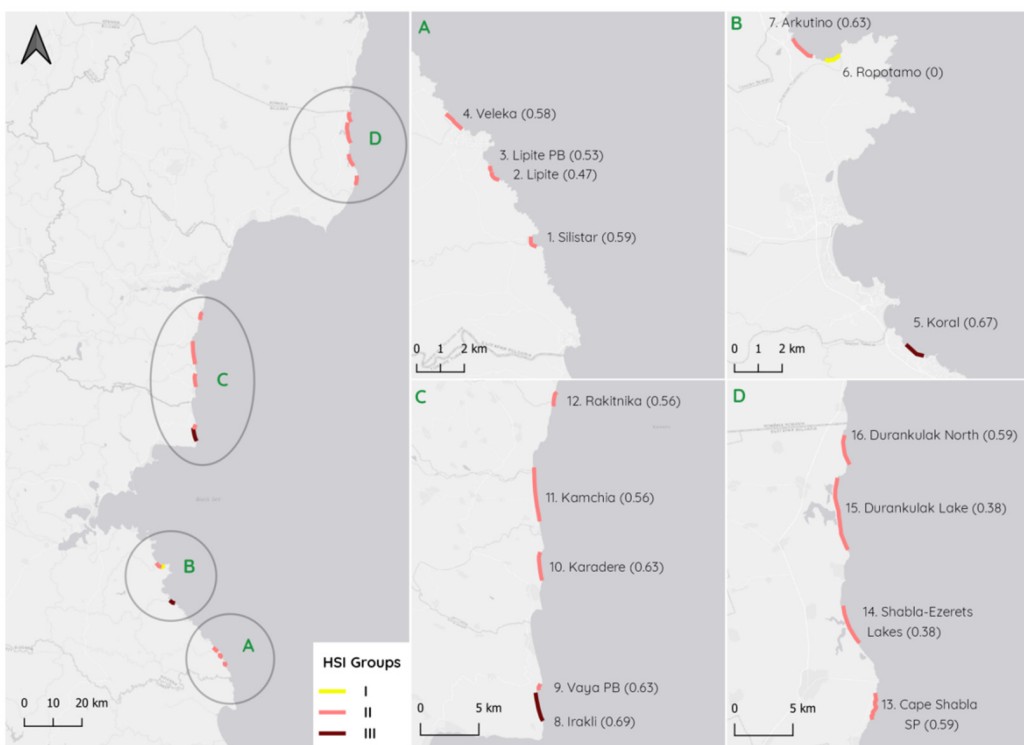

**Figure 14.** HSI values along with corresponding sensitive group. Global view on the left map and zooms of (**A**–**D**) zones.

**Table 7.** Site scores for HSI parameters. * Site belonging to a Strict Nature Reserve (IUCN; Ia) and accessible by a walk >45 min (Category I site; no further investigation is required). ** For sites located in two municipalities, the highest value of HSI was considered.

| Site | Province | Municipality | Category | Access | PAMC | TIR | PD | Beach Typology | HI | Beds | Population | HSI | Group |
|---|---|---|---|---|---|---|---|---|---|---|---|---|---|
| 1. Silistar | | Tsarevo | II | 4 | 3 | 5 | 1 | | 0.58 | 4 | 3 | 0.59 | II |
| 2. Lipite | | Tsarevo | II | 2 | 3 | 5 | 1 | | 0.42 | 4 | 3 | 0.47 | II |
| 3. Lipite PB | | Tsarevo | II | 3 | 3 | 5 | 1 | | 0.50 | 4 | 3 | 0.53 | II |
| 4. Veleka | | Tsarevo | III | 4 | 3 | 5 | 1 | 3 | 0.58 | 4 | 3 | 0.58 | II |
| 5. Koral | Burgas | Tsarevo | III | 3 | 5 | 5 | 1 | 3 | 0.67 | 4 | 3 | 0.67 | III |
| 6. Ropotamo * | | Primorsko | I | | | | | | | | | | I |
| 7. Arkutino | | Primorsko | II | 3 | 4 | 5 | 1 | | 0.58 | 5 | 3 | 0.63 | II |
| 8. Irakli | | Nesebar | II | 3 | 4 | 5 | 2 | | 0.63 | 4 | 5 | 0.69 | III |
| 9. Vaya PB | | Nesebar | II | 3 | 4 | 5 | 2 | | 0.63 | 4 | 5 | 0.63 | II |
| 10. Karadere | | Byala | II | 3 | 4 | 5 | 1 | | 0.58 | 5 | 3 | 0.63 | II |
| 11. Kamchia ** | | Dolni Chiflik | II | 3 | 4 | 2 | 3 | | 0.54 | 3 | 2 | 0.50 | II |
| | Varna | Avren | II | | | 3 | 1 | | 0.50 | 4 | 4 | 0.56 | II |
| 12. Rakitnika ** | | Avren | II | 4 | 3 | 3 | 1 | | 0.50 | 4 | 4 | 0.56 | II |
| | | Varna | II | | | 2 | 5 | | 0.63 | 1 | 4 | 0.56 | II |
| 13. Shabla–Ezerets SP | | Shabla | III | 5 | 4 | 3 | 1 | 3 | 0.67 | 5 | 1 | 0.59 | II |
| 14. Shabla–Ezerets Lakes | | Shabla | II | 2 | 3 | 3 | 1 | | 0.33 | 5 | 1 | 0.38 | II |
| 15. Durankulak Lake | Dobrich | Shabla | II | 2 | 3 | 3 | 1 | | 0.33 | 5 | 1 | 0.38 | II |
| 16. Durankulak North | | Shabla | II | 3 | 4 | 3 | 1 | | 0.50 | 5 | 1 | 0.50 | II |

(1) **Access difficulty** is an essential component of management approaches to regulate and protect sites from too many tourists. Among the 16 investigated sectors, only three sites were easily accessible by a <10-min walk from the nearest car park, i.e., Cape Shabla SP (rated 5, Table 7), Rakitnika and Veleka (both rated 4), and eight required a 10–25 min promenade (rated 3). Lower scoring was noticed for Lipite, Irakli, Shabla Ezerets Lakes and Durankulak Lake, which demanded at least 25 min of walking (rated 2, Table 7).

(2) **Protected Area Management Category** was assessed accordingly to the standard methodology provided by the International Union for Conservation of Nature (IUCN) [75], ranging from protected areas very strictly managed, e.g., Ropotamo Strict Nature Reserve (Ia), to ones managed in a relatively permissive way, e.g., Silistar Protected Area (VI). As shown in Table 5, sectors were partially or completely covered by several national and international designations, e.g., Nature Parks, Natura 2000, apart from Koral beach (rated 5, Table 7). All sites belonged to the Natura 2000 network characterized by 26 Marine Protected Areas (MPAs), with most including a coastal land area with only a narrow strip protruding into the sea, 11 SPAs (under the Birds Directive), 13 SCIs (under Habitats Directives) and two SCI–SPAs under both directives [51]. However, the practical application of Natura 2000 still poses major problems, since its process of implementation is coordinated and managed by the Ministry of Environment and Water, while CDDA (Nationally designated areas) is managed by different Institutions. Today, there is still a lack of approved and operational management plans for coastal protected areas and MPAs [51]. Because of a lower grade of protection, sites located within State Game Husbandries combined with MPAs obtained a rating of 4, i.e., Vaya PB, Arkutino or Cape Shabla (Table 7), whilst sites within a Nature Park (Category V, IUCN), i.e., Strandzha, Protected Area/Site (VI), e.g., Silistar, Veleka and Shablensko Ezero, gave intermediated values (3) (Table 7). At Kamchia (rated 4), the eponymous Protected Site (VI) and Strict Nature Reserve (Ia) (Table 5), were not considered, as they only related to the river outlet and not the beach. Irakli is a similar case, as a Natural Monument area (III) situated along the southern sector was not reflected in its rating (3), since it only represents a minor part of the total beach length.

(3) **Tourism Intensity Rate (TIR) and Population Density (PD)** were evaluated by using the dataset provided by the Ministry of Tourism [76] and the National Statistical Institute [77] (2021), both at municipality scale (Nomenclature of Territorial Units for Statistics, NUTS 5), given that provincial averages bear the risk of misleading disparities. Top grades for TIR (5) were registered at several municipalities (Table 7), suggesting that tourist capacity is superior to that of the permanent population. Highest values were noticed at Primorsko, i.e., 4346 tourist beds per 1000 inhabitants (2020), Nessebar (3216 beds per 1000 inhabitants), Tsarevo (1757 beds) and Byala (1088 beds), whilst Avren and Varna showed the lowest ratings (<30% beds per inhabitants; rated 2, Table 7). However, an opposite trend was recorded for PD. Bulgaria is experiencing a decline in population, which began at the beginning of the 1990s, and currently is losing roughly around 50,000 citizens per year [78], this being one of the major issues/challenges that the governing authority has to deal with. With regard to coastal municipalities, low values were commonly observed (≤2), except at Varna (1438 inhabitants/km$^2$; rated 5, Table 7) and, to a lesser extent, at Dolni Chiflik, with 152 inhabitants per km$^2$ (rated 3, Table 7). Lastly, in the case of Kamchia and Rakitnika, since both sectors belonged to two different municipalities, the highest values obtained for the TIR and PD were chosen for the HI assessment.

A Correction Factor value (CF) was then calculated for each site by considering the following variables, once again, at NUTS 5.

(1) **Evolution of tourist beds,** obtained from the Ministry of Tourism [76], was generally characterized by high values (≥4) for most sites (Table 7). Only two municipalities presented lower scores, respectively, Dolni Chiflik (rated 3, Table 7), with an increase of 44% during the period 2006–2021, and Varna, which stands out from the rest with a

10% decrease (rated 1, Table 7). Opposite results were noticed at Shabla and Byala, both rated 5, with, respectively, an increase of 588% (538 in 2006, and 3702 in 2021) and 434% (from 292 to 1561).

(2)  **Evolution of the resident population** was also considered to complement the latter variable, since a decrease/increase of the resident population can also have a significant impact on coastal areas. In this case, and considering the current Bulgarian situation, a stable evolution was reflected by intermediate scores (3), whereas an increase >25% obtained the top rating (5), and vice versa, for a decrease >25% (1) (Table 7). The municipality of Nessebar (rated 5) recorded a 42% rise in inhabitants from 2005 (20,938) to 2020 (29,814) [77]. A slight increase was also registered at Varna (rated 4, Table 7). The lowest rate corresponded to Shabla, which showed a 27% decrease, with 5959 inhabitants in 2005 and 4337 in 2020. Finally, Tsarevo, Primorsko and Byala municipalities maintained a stable population in the last 15 years (rated 3, Table 7).

As stated for natural systems, a Human Sensitivity Index (HSI) was obtained by linking values obtained for HI and CF established by Mooser et al. [37] (Table 7 and Figure 15).

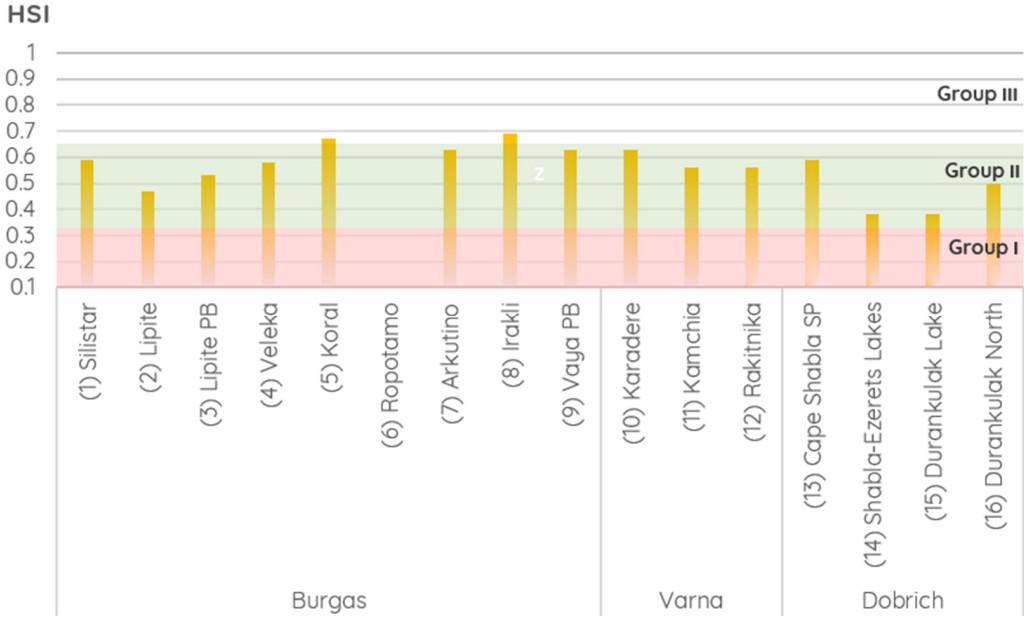

**Figure 15.** Site scores obtained for the Human Sensitivity Index (HSI).

Finally, a Total Sensitivity Index (TSI) was estimated by associating the values formerly achieved for NSI and HSI. Scores are presented in Table 5 and Figure 16. High values of TSI enabled us to identify/highlight sites that are very sensitive to both natural processes and human impacts. Only two sectors were included in the Group I, i.e., Ropotamo and Cap Shabla SP (both 0.32). The sites mostly belonged to Group II (9), but five fell within Group III, being broadly exposed to both factors: Silistar (0.72), Veleka (0.70), Irakli (0.68), Vaya PB (0.76) and Karadere (0.78).

*4.3. Beauty versus Sensitivity: Priorities in Terms of Management*

The above lead to the following question: which scenarios are the most sensitive to human and/or natural processes among the top scenic sites? To answer this, the relationship between NSI/HSI (CSSI) and "D" value (CSES) was analyzed below, with the aim of making data easier to read and interpret in order to identify priorities in terms of management (Figure 16). Our comparison of NSI and HSI clearly shows that investigated sectors were substantially much more exposed to natural processes than human pressure (Figure 16A). This is because investigated beach typologies were predominantly remote, and their overall

assessment was considerably lowered by the "Population Density" parameter. Limits of Class I (D: 0.85; CSES) and Group III (SI: 0.66) were signaled in Figure 16B–D to identify the most attractive and sensitive sites.

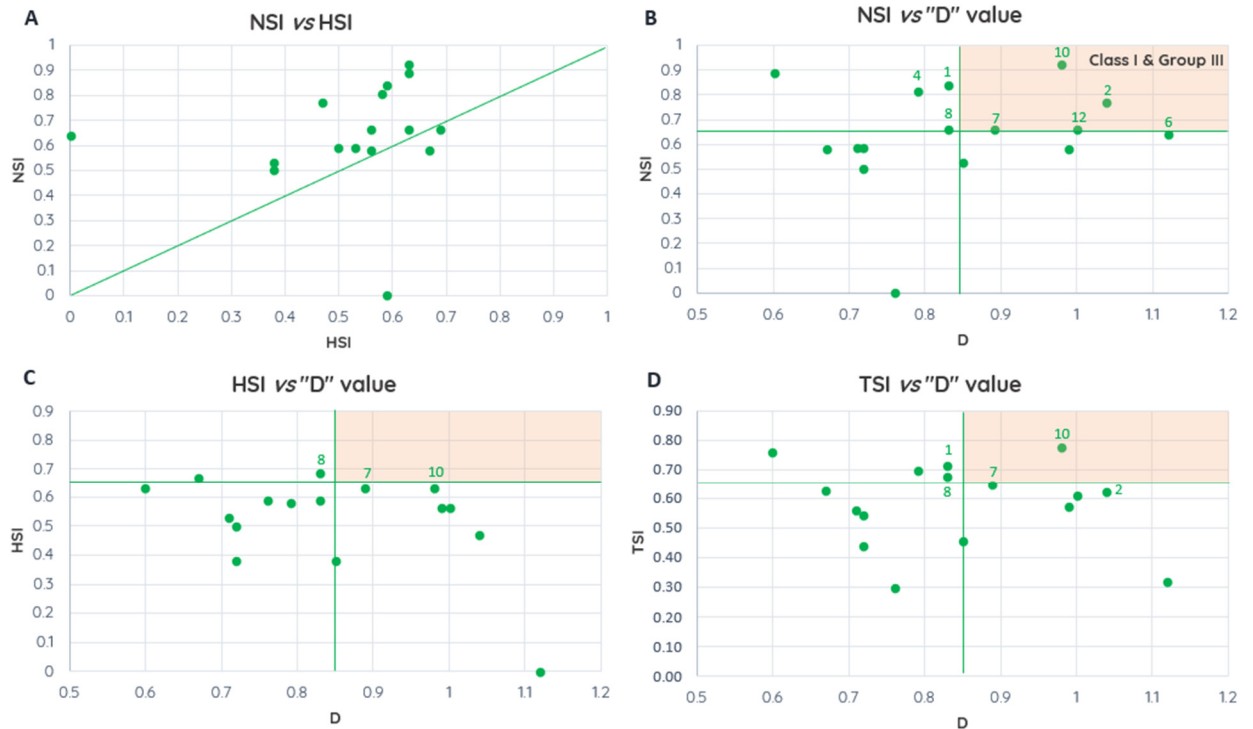

**Figure 16.** CSES versus CSSI: Natural Sensitivity Index (NSI) versus Human Sensitivity Index (HSI) (**A**), NSI versus "D" value (CSES) (**B**), HSI versus "D" value (**C**) and Total Sensitivity Index (TSI) versus "D" value (**D**), with corresponding location map number for highly sensitive sites of great scenic value.

4.3.1. NSI versus "D" Value

Only one site belonged to Group I (Cape Shabla SP; Category I) and seven to Group II, whilst half (8) were considered as very sensitive to natural processes (NSI ≥ 0.66; Group III). The most sensitive sceneries were Karadere (NSI: 0.92) and Vaya PB (NSI: 0.89), as both recorded the highest values for EI parameters: dry beach < ICZ (5), fine sand (5) and absence of shore platform (5) (Table 6). For either, very few sustainable measures could be carried out, since both are predominantly surrounded by cliff formations.

Four sites belonged to Class I and Group III and, thus, require specific attention from managers (Figure 16B): Arkutino (point 7, Figure 16B), Rakitnika (12), Karadere (10) and Lipite (2). With regard to Rakinika (12), one of the most attractive places (D: 1.00), it would be essential to control illegal access of vehicles, mainly four-wheel-drive vehicles, to preserve the dune system, together with beach users' security. From a scenic approach, the beach of Arkutino was divided into two different sectors during the field surveys: a recreational one (not investigated because of its low scenic value) and a "natural" one, 1 km in length (Table 5). The natural sector, despite showing a strong and resilient dune complex (rated 1; Table 6), recorded very high erosion rates (≤ICZ; rated 5) associated with the presence of a breakwater in its northern limit. Regarding Lipite, apart from artificial beach nourishment, very few effective measures could be carried out to reduce its sensitivity, as Lipite and Karadere are not able to migrate landward because they are backed by cliffs and bluffs.

However, a few places included in Group III and close to the limits of Class I could upgrade/strengthen their attractiveness and resilience by limiting or avoiding human trampling on dunes and recreational activities on the beach (e.g., bars, etc.), i.e., Silistar (1) and

Veleka (4). With these interventions, both sites could be upgraded to Class I, accomplishing, in this way, two objectives with one action.

### 4.3.2. HSI versus "D" Value

Most investigated sectors belonged to Group II (13), and no sites apart from Ropotamo, initially included in Category I, were considered as "not sensitive" (Group I) (Figures 14 and 15). Within Group II, the lowest values were found at Durankulak and Shabla–Ezerets Lakes (both 0.38; Table 7 and Figures 14 and 15), because of their restrictive access (2), intermediate values for PAMC (3), very weak density of population (1) and a decline of this latter counterbalancing a significant increase of tourist beds (5) (Table 7). In a similar case, Lipite showed low scores for HI parameters but higher values related to CF, slightly raising the final HSI value to 0.47. Regarding the most sensitive sceneries, two places stood out from the rest: Koral and Irakli (Group III), both located in Burgas province (Figures 14 and 15). At Koral, high grades observed for PAMC (rated 5) and TIR, corresponding to the municipality of Tsarevo (1757 beds/1000 inhabitants; rated 5, Table 7), lead the site to Group III, with a HSI value of 0.67. The Koral HSI value could be reduced to 0.58, thus downgrading it to Group II, by securing this sector within an adequate PAMC.

No site belonged to Class I and Group III, but three sites seemed to need more management involvement: Arkutino (point 7, Figure 16C), Irakli (8) and Karadere (10). For the first two abovementioned sites, management interventions should be required to improve their Protected Area Management Category, as they only belong to State Game Husbandries and Natura 2000 SCI and SPA. Furthermore, it would be interesting to extend the current Irakli Protected Area (VI) toward the northern coastal sector, thereby including most of the beach. Nessebar municipality (where belongs Irakli) also recorded the highest increase of locals and registered an increase in tourist beds of 57% (period 2005–2020), which corresponded to a capacity three times above the permanent population. Concerning Karadere, high scores were also recorded for CF parameters, and particularly at tourist beds, i.e., an increase of 434% in the last 15 years corresponding to the municipality of Byala (rated 5; Table 7). Karadere is also the only place with a TSI and "D" value falling into both categories (Figure 16D).

Likewise, it would be convenient to determine the tourism carrying capacity of several destinations, i.e., Tsarevo, Nessebar, Primorsko and Byala (Table 7), which present a tourism flux widely superior to the number of its resident population. Local management systems are generally not prepared to manage the associated environmental stress, e.g., the insufficiency of wastewater treatment plants, etc., resulting in even greater damages for nearby ecosystems [49]. Local municipalities need to understand the carrying capacity of each site and ways to spread tourists over a greater area, e.g., by introducing marketing strategies to promote "Local jewels" to reach a large public and attract new types of tourists. This can be performed by using coastal scenic beauty as a qualitative resource, i.e., Class I sites (CSES), within the aims of the ECOTOUR-NET project framework (Development of the ecotourism network in the Black Sea region) [79] funded by the "Joint Operational Program Black Sea region 2014–2020". Some countries have already proved how important it is to use labels as a marketing brand that indicates quality, e.g., in the UK, with the eco-label "Areas of Outstanding Natural Beauty" (AONB) [80].

### 5. Conclusions

This paper is a contribution focused on the preservation/enhancement of the natural scenic beauty of investigated coastal areas by providing the following:

(i)   The characterization of most attractive coastal scenic sites and associated weakness and sensitivity to natural and human induced factors,

(ii)  The promotion of their potential development under ecotourism principles.

Bulgaria offers an impressive scenic diversity along a limited coastline length (432 km), having unique places, such as Strandzha, with a remarkable oak forest dating from the Tertiary. Seven sites were classified as "extremely attractive with outstanding features"

(Class I), but with slight management efforts, several Class II sites could efficiently improve their attractiveness and be upgraded to Class I. The results showed how litter generally downgraded the scenic value of sites. Emphasis should be also devoted to reducing intrusive "Utilities", essentially linked to recreational activities, as observed at Silistar and Veleka. Further, it is fundamental to manage each beach according to its typology, bearing in mind that scenic quality has to prevail over recreational services in remote areas.

This paper also reveals that investigated sites were generally more sensitive to environmental impacts than human pressure, and half were categorized as very sensitive to natural processes (NSI; Group III). In this regard, special attention should be paid to protect dune systems by reducing the illegal circulation of vehicles that leads to the loss of biodiversity, dune erosion and fragmentation at places linked to the enhancement of existing washover fans. By mitigating bad human practices and uses, lots of coastal areas would jointly increase their resilience to erosion in a climate-change context with critical predictions of RSLR in order to ensure the safety of beach users and improve their scenic attractiveness. Therefore, it is of the utmost importance to strengthen areas with low (or no) grades of protection by extending existing protected areas, e.g., Irakli, Karadere, Kamchia, or by creating new ones, i.e., Koral. Another concern is the inability to implant management plans to ensure the sustainable conservation of Natura 2000 sites. The growing tourist capacities of Byala, Primorsko and Nessebar should be also controlled in order to avoid future scenarios of overcrowding and related adverse effects, e.g., landscape and environmental degradation, uncontrolled mass tourism, loss of quality in services provided to visitors and loss of its positive image as a pleasant tourist destination.

Finally, the results obtained in this paper could be used as a baseline for the establishment of a novelty "coastal scenic award" to (i) promote extremely attractive sites along the Bulgarian coast under the umbrella of sustainable tourism, e.g., Class I sites; and (ii) increase the interest of local managers in landscape preservation, within the scope of the ECOTOUR-NET project.

**Author Contributions:** A.M. and G.A. designed the study and participated in all phases. H.S. and M.S. participated in the development of results/discussion and provided specific information related to the study area, i.e., erosion rates, categories of protection and tourism trends at municipality scale. A.T.W. provided a global structural discussion and made English corrections. P.P.C.A. made contributions regarding the conceptual approach (ideas, formulation of research goals) and provision of resources (materials, literature review). A.M., H.S. and M.S. carried out the field work observations. All authors have read and agreed to the published version of the manuscript.

**Funding:** This research was partially funded by Università degli Studi di Napoli Parthenope (D.R. 890/19) as the first author was supported by a PhD scholarship under the program "Environmental Phenomena and Risk", cycle 35th.

**Acknowledgments:** Thanks to the Center for Coastal Marine Studies (CCMS, Varna, Bulgaria) for the strong support given during the research stay (2021). This is a contribution to the PROPLAYAS Network and the Andalusia PAI Research Group RNM-328 (Spain).

**Conflicts of Interest:** The authors declare no conflict of interest.

## Appendix A

**Table A1.** Equations used for the assessment of EI, NSI, HI, HSI, TSI and Correction Factors (natural and human; CSSI method).

| Indexes and Categories | Equations | Parameters |
|---|---|---|
| Erodibility Index (1) for Category II sites ($EI_{C2}$) | $EI_{C2} = E_{BF} = \dfrac{\frac{Pn_1 + Pn_2 + \frac{Pn_{3a} + Pn_{3b}}{2}}{n_{Pn}} - 1}{A - 1}$ | $E_{BF}$: erodibility of beach face parameters $Pn$ : natural parameter $Pn_1$ : dry beach evolution $Pn_2$ : sediment grain size $Pn_{3a}$ : rocky shore width $Pn_{3b}$: rocky shore location $n_{Pn}$: number of natural parameters (3) $A$: maximum attribute value (5) |
| Erodibility Index (2) for Category III sites ($EI_{C3}$) | $EI_{C3} = E_{BF} \times \frac{2}{3} + E_{DS} \times \frac{1}{3}$ | $E_{DS}$: erodibility of dune system parameters |
| Erodibility of dune system (3) ($E_{DS}$) | $E_{DS} = \dfrac{\frac{Pn_4 + Pn_5 + Pn_6 + Pn_7}{n_{Pn}} - 1}{A - 1}$ | $Pn_4$: dune height $Pn_5$ : dune width $Pn_6$: vegetation cover $Pn_7$: washovers |
| Natural Correction Factor (4) ($CF_N$) | $CF_N = \dfrac{\frac{\frac{c_{1a}\ c_{1b}}{2} + c_2 + c_3 + c_4}{n_C} - 1}{A - 1}$ | $c_{1a}$: significant wave height $c_{1b}$: angle of wave approach $c_2$: tidal range $c_3$: sea-level rise $c_4$: storm surge |
| Natural Sensitivity Index (5) (*NSI*) | $NSI = EI \times \frac{3}{4} + CF_N \times \frac{1}{4}$ | |
| Human Impact Index (6) for Category II sites ($HI_{C2}$) | $HI_{C2} = \dfrac{\frac{Ph_1 + Ph_2 + \frac{Ph_{3a} + Ph_{3b}}{2}}{n_{Ph}} - 1}{A - 1}$ | $Ph$: human parameter $Ph_1$: access difficulty $Ph_2$: protected area management category $Ph_{3a}$: tourism intensity rate $Ph_{3b}$: population density $n_{Ph}$: number of human parameters $A$: maximum attribute value (5) |
| Human Impact Index (7) for Category III sites ($HI_{C3}$) | $HI_{C3} = \dfrac{\frac{Ph_1 + Ph_2 + \frac{Ph_{3a} + Ph_{3b}}{2} + Ph_4}{n_{Ph}} - 1}{A - 1}$ | $Ph_4$: beach typology |
| Human Correction Factor (8) ($CF_H$) | $CF_H = \dfrac{\frac{c_1 + c_2}{n_C} - 1}{A - 1}$ | $c_1$: tourism trend $c_2$: population trend |
| Human Sensitivity Index (9) (*HSI*) | $HSI = HI \times \frac{3}{4} + CF_H \times \frac{1}{4}$ | |
| Total Sensitivity Index (10) (*TSI*) | $TSI = \frac{NSI + HSI}{2}$ | |

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
