# Peer review of "Most Attractive Scenic Sites of the Bulgarian Black Sea Coast: Characterization and Sensitivity to Natural and Human Factors"

_land, doi:10.3390/land11010070_

Round 1
Reviewer 1 Report
Present manuscript is enlargement and application of model presented in Water 2021, 13, 49.
Complex evaluation of coastal areas is presented using methodology already published and no theoretical results are presented, but the power of the manuscript is in application sphere. I think that this is important survey that deserves publishing, however especially according to the selection of localities its results seem like to be “self-fulfilling prophecy”.
I am talking about methodology – especially site selection for the survey:
lines 182-186: “Land cover viewers and satellite images were used to 182 give a first approximation on the location of most beautiful natural coastal areas despite 183 them being located or not in protected areas. The images were basically used to eliminate 184 urbanized areas and select places, mostly beaches, which appeared of great scenic values 185 because the presence of dunes/cliffs and no human activities/impacts. Sites accessible by 186 a walk longer than1.5 h were omitted“
- such selection is extremely subjective, on the other hand the methods used for evaluation are presented as objective as can be
- question is – only those 16 localities were select by this process? No others?
It is not clear from methods how weights were calculated.
It is also disputable why indexes were calculated and in the final step the TSI is calculated as simple arithmetic mean of NSI and HIS – I do not think that this number has any importance for the aim of the paper.
Please make clear, why indexes were calculated – we know that changes are following gradients and when many indicators were used, some multivariate analysis could be appropriate to solve this issue.
The manuscript could be accepted only if these remarks are solved in the text.
Author Response
Thank you for your observations.
Response to reviewer # 1
Present manuscript is enlargement and application of model presented in Water 2021, 13, 49.
Complex evaluation of coastal areas is presented using methodology already published and no theoretical results are presented, but the power of the manuscript is in application sphere. I think that this is important survey that deserves publishing, however especially according to the selection of localities its results seem like to be “self-fulfilling prophecy”.
I am talking about methodology – especially site selection for the survey:
lines 182-186: “Land cover viewers and satellite images were used to give a first approximation on the location of most beautiful natural coastal areas despite them being located or not in protected areas. The images were basically used to eliminate urbanized areas and select places, mostly beaches, which appeared of great scenic values because of the presence of dunes/cliffs and no human activities/impacts. Sites accessible by a walk longer than1.5 h were omitted“
- such selection is extremely subjective, on the other hand the methods used for evaluation are presented as objective as can be ‘mostly beaches, which appeared of great scenic values because of the presence of dunes/cliffs and no human activities/impacts. Sites accessible by a walk longer than1.5 h were omitted.’
Response: Thank you for your observation, we agree with you: this part was not sufficiently and properly explained in the methods’ section. We completely rephrased it. We hope it is clearer now, please see lines 183-223.
It is not clear from methods how weights were calculated
Response: For coastal scenic beauty, weights were presented in Table 1. Title of Table 1 has been revised. For coastal sensitivity, a new table in an Annex I was added, please see Table A1.
It is also disputable why indexes were calculated and in the final step the TSI is calculated as simple arithmetic mean of NSI and HIS – I do not think that this number has any importance for the aim of the paper.
Response: High values of TSI enable to identify/stand out sites very sensitive to both natural processes and human impacts.
Please make clear, why indexes were calculated – we know that changes are following gradients and when many indicators were used, some multivariate analysis could be appropriate to solve this issue.
Response: We are sorry we do not understand the question, the evaluation index “D” is the manner to quantify scenic beauty as well as NSI and HSI make possible the quantification of scenic sensitivity.
Reviewer 2 Report
This is an interesting study on beach assessment. The research questions are well identified. The research methodology is well developed. The assessment results are sufficiently presented. However, there are issues that need to be addressed before the acceptance for publication.
First, the authors need to add a Discussion section to present the implications of the assessment results for beach management.
Second, the abstract is loosely written and needs a rewriting to make it more precise.
Third, the use of the phrases “human occupation” and “coastal occupation” is very confusing. Based on your writing, it means different things in different context such as coastal development, coastal population and etc. Please replace them with phrases having precise meanings that are commonly used in literature.
Four, there are many language issues that need to be addressed. Below are just a few examples. Please have someone carefully copy-edited the manuscript.
- L20: should be “climate change” instead of “Climate Change” and “Sea, Sun and Sand” instead of “Sea, Sun and Sea”.
- L28: Use “mistakes” instead of “errors”.
- L50: Change the word for “Occupation”.
- L59: no need to capitalize the first letter on climate change.
- L64: delete “assuming an optimal cost-benefit analysis”.
- L70: “occupation” again?
- L72: no “s” after the word “Infrastructure”.
- L74: Replace “Soil” with “Land”.
- L78: no “s” after the word “population”.
- L191-197: no sure which style of citation you want to use. Please use on and be consistent.
Author Response
Thank you for your observations.
Response to reviewer # 2
This is an interesting study on beach assessment. The research questions are well identified. The research methodology is well developed. The assessment results are sufficiently presented. However, there are issues that need to be addressed before the acceptance for publication.
First, the authors need to add a Discussion section to present the implications of the assessment results for beach management. You may refer to the above article to see how different strategies be used to address the different threats.
Response: Thank you for your observations and suggestions.
We are sorry, it was not clear to us if the reviewer refers to a specific paper or not (we did not find any reference/citation in the revision or attached document).
From one side we perfectly understand that, generally, a paper in which results and discussion are separated appears to be more “elegant” and “well-organized” respect to one in which results and discussion are combined. From the other side, we guess that it is not the case because several complex and different aspects are exposed and discussed in this paper. For this reason we guess that it is better to maintain results and discussion united, since we are very convinced that it would be very unwieldy and difficult for the reader to follow the meaning and logic of the paper if results about scenic beauty and sensitivity to natural processes and human impacts are firstly presented and, than, a discussion of each one of such aspects is presented. Further, the structure used in this paper follows the one used in a similar paper on the scenic beauty of different sites in the Balearics Islands (Spain), published this year in LAND.
Second, the abstract is loosely written and needs a rewriting to make it more precise.
Response: thank you, we had re-written the abstract and added new information.
Third, the use of the phrases “human occupation” and “coastal occupation” is very confusing. Based on your writing, it means different things in different context such as coastal development, coastal population and etc. Please replace them with phrases having precise meanings that are commonly used in literature.
Response: thank you very much, we revised all terms and made changes (see text).
Four, there are many language issues that need to be addressed. Below are just a few examples. Please have someone carefully copy-edited the manuscript.
Response: thank you very much for your work; we carried out all suggested changes and others proposed by a native English speaking coauthor (A.T. W.).
Round 2
Reviewer 1 Report
Thank you for new version of manuscript and the changes made.
Author Response
Thank you!!
Reviewer 2 Report
I still feel a separate Discussion section on the implication of the results on beach management in term of how to mitigate the negative impacts of natural hazards and human development the would be helpful to the readers.
Author Response
Dear Reviewer
according to your suggestions we created a new session "4.3 Beauty versus Sensitivity: priorities in term of management. All changes along with the text are marked in red.
Thank you! regards.